# TrpML-mediated astrocyte microdomain Ca$^{2+}$ transients regulate astrocyte–tracheal interactions

**Zhiguo Ma\*, Marc R Freeman\***

Vollum Institute, Oregon Health and Science University, Portland, United States

**Abstract** Astrocytes exhibit spatially-restricted near-membrane microdomain Ca$^{2+}$ transients in their fine processes. How these transients are generated and regulate brain function in vivo remains unclear. Here we show that *Drosophila* astrocytes exhibit spontaneous, activity-independent microdomain Ca$^{2+}$ transients in their fine processes. Astrocyte microdomain Ca$^{2+}$ transients are mediated by the TRP channel TrpML, stimulated by reactive oxygen species (ROS), and can be enhanced in frequency by the neurotransmitter tyramine via the TyrRII receptor. Interestingly, many astrocyte microdomain Ca$^{2+}$ transients are closely associated with tracheal elements, which dynamically extend filopodia throughout the central nervous system (CNS) to deliver O$_2$ and regulate gas exchange. Many astrocyte microdomain Ca$^{2+}$ transients are spatio-temporally correlated with the initiation of tracheal filopodial retraction. Loss of TrpML leads to increased tracheal filopodial numbers, growth, and increased CNS ROS. We propose that local ROS production can activate astrocyte microdomain Ca$^{2+}$ transients through TrpML, and that a subset of these microdomain transients promotes tracheal filopodial retraction and in turn modulate CNS gas exchange.

**\*For correspondence:**
mzh@ohsu.edu (ZM);
freemmar@ohsu.edu (MRF)

**Competing interests:** The authors declare that no competing interests exist.

## Introduction

Astrocytes exhibit two major types of Ca$^{2+}$ signaling events, whole-cell fluctuations and near-membrane microdomain Ca$^{2+}$ transients (*Khakh and McCarthy, 2015*). Whole-cell transients are coordinated across astrocyte networks and regulated by adrenergic receptor signaling. Emerging data suggests these transients are important for state-dependent changes (*Ding et al., 2013*; *Ma et al., 2016*; *Paukert et al., 2014*; *Srinivasan et al., 2015*), and involve TRPA1 channels that regulate the insertion of neurotransmitter transporters like GAT-3 into astrocyte membranes to alter neurophysiology (*Shigetomi et al., 2011*). Whole-cell astrocyte Ca$^{2+}$ transients in the *Drosophila* CNS are also stimulated by the invertebrate equivalents of adrenergic transmitters, octopamine (Oct) and tyramine (Tyr). Octopamine and tyramine stimulate cell-wide astrocyte Ca$^{2+}$ increase through the dual-specificity Octopamine-Tyramine Receptor (Oct-TyrR) and the TRP channel Water witch (Wtrw). This astrocyte-mediated signaling event downstream of octopamine and tyramine is critical for in vivo neuromodulation: astrocyte-specific elimination of Oct-TyrR or Wtrw in larvae blocks the ability of octopamine and tyramine to silence downstream dopaminergic neurons, and alters both simple chemosensory behavior and a touch-induced startle response (*Ma et al., 2016*). Adrenergic regulation of whole-cell astrocyte Ca$^{2+}$ transients is therefore an ancient and broadly conserved feature of metazoan astrocytes.

The mechanisms that generate astrocyte microdomain Ca$^{2+}$ transients are not understood, nor are the precise in vivo roles for this type of astrocyte signaling (*Bazargani and Attwell, 2016*; *Khakh and McCarthy, 2015*). In mammals, astrocyte microdomain Ca$^{2+}$ transients occur spontaneously, do not require neuronal activity (*Nett et al., 2002*), depend on extracellular Ca$^{2+}$ (*Rungta et al., 2016*; *Srinivasan et al., 2015*), and persist in cultured astrocytes, which has been

used to argue they are cell-autonomous (*Khakh and McCarthy, 2015*; *Nett et al., 2002*). A recent study described close association of astrocyte microdomain Ca$^{2+}$ transients with mitochondria, and found that pharmacological blockade of the mPTP led to a suppression of transients, while ROS led to an enhancement of transients. These observations led to the proposal that astrocyte microdomain Ca$^{2+}$ transients were generated by opening of the mPTP during oxidative phosphorylation, perhaps as a means to balance mitochondrial function and ongoing astrocyte support with local metabolic needs (*Agarwal et al., 2017*).

In this study, we report that *Drosophila* astrocytes exhibit spontaneous, activity-independent microdomain Ca$^{2+}$ transients, and show they are mediated by the TRP channel TrpML. Many astrocyte microdomain Ca$^{2+}$ transients are associated with branches and filopodia extending from CNS trachea, an interconnected set of tubules that allow for gas exchange in the larval CNS, and astrocyte microdomain Ca$^{2+}$ transients precede the onset of filopodial retraction. Astrocyte microdomain Ca$^{2+}$ transients are regulated by ROS and loss of TrpML leads to tracheal overgrowth and increased CNS ROS. We propose that one in vivo role for tracheal–astrocyte interactions is to regulate CNS gas exchange, with tracheal filopodia-dependent local hyperoxia resulting in increased production of ROS, which gates TrpML to generate local astrocyte microdomain Ca$^{2+}$ transients, ultimately promoting tracheal retraction and reducing local O$_2$ delivery.

## Results

### *Drosophila* astrocytes exhibit microdomain Ca$^{2+}$ transients

To monitor the near-membrane Ca$^{2+}$ activity in astrocytes, we expressed myristoylated GCaMP5a (myr-GCaMP5a) in astrocytes using the astrocyte-specific *alrm-Gal4* driver. We acutely dissected 3$^{rd}$ instar larval CNS and live-imaged myr-GCaMP5a signals in the ventral nerve cord (VNC) (*Ma et al., 2016*). We collected images at the midpoint of the neuropil along the dorsoventral axis for 6 min time windows (*Figure 1A*). We found that astrocyte microdomain Ca$^{2+}$ transients exhibited diverse waveforms, with variable durations and frequencies (*Figure 1A*; *Figure 1—video 1*). The average full width at half maximum (FWHM) for these Ca$^{2+}$ transients was 5.5 ± 2.26 (mean ± SD) seconds (*Figure 1B*). Microdomain Ca$^{2+}$ transients frequently occurred at the same location, suggesting there are hotspots where microdomains repeatedly occur for a given astrocyte. The majority of foci exhibited 1–3 events during the 6 min imaging window (*Figure 1C*), and Ca$^{2+}$ transients at different sites did not exhibit obvious synchrony with one another. We also observed microdomain Ca$^{2+}$ transients of a similar rise-and-fall pattern, although with a slightly shorter duration (FWHM, 1.7 ± 0.08 s, mean ± SD) in the astrocytes of intact L1 (1$^{st}$ instar) larvae (*Figure 1- Figure 1—figure supplement 1A and B*; *Figure 1—video 2*), suggesting the microdomain Ca$^{2+}$ transients in the acute CNS preparations we observed largely reflect in vivo astrocyte activity, although with some differences in duration. Blockade of action potential firing with tetrodotoxin did not alter astrocyte microdomain Ca$^{2+}$ transients, although they were eliminated by removal of extracellular Ca$^{2+}$ and were sensitive to the Ca$^{2+}$ channel blocker lanthanum chloride (LaCl$_3$) (*Figure 1D*), suggesting Ca$^{2+}$ entry from extracellular space is essential for generation of astrocyte microdomain Ca$^{2+}$ transients.

Astrocytes tile with one another and occupy unique spatial domains in the CNS. We sought to determine whether microdomain Ca$^{2+}$ transients spanned astrocyte-astrocyte cell boundaries, or if they appeared only within the domain of single cells. We used a flippable construct expressing either QF or Gal4 under the control of the *alrm* promoter, along with two genetically encoded Ca$^{2+}$ indicators: *QUAS::myr-GCaMP5a* and *UAS::myr-R-GECO1* (*Figure 1—figure supplement 1 - Figure 1C*). To confirm both myr-GCaMP5a and myr-R-GECO1 behaved similarly, we first examined double-positive cells and found both can detect the same microdomain Ca$^{2+}$ transients (*Figure 1—figure supplement 1 - Figure 1D*), and in cells exclusively expressing one of these two Ca$^{2+}$ indicators there were no differences in the overall frequency of the microdomain Ca$^{2+}$ events detected between these sensors (*Figure 1E*). We then identified cell boundaries between myr-GCaMP5a/myr-R-GECO1 single-labeled cells, and examined the dynamics of astrocyte microdomains across those boundaries. We observed coincident signaling with myr-GCaMP5a and myr-R-GECO1 (*Figure 1E*). These data indicate that individual astrocyte microdomain Ca$^{2+}$ transients can span astrocyte-astrocyte borders. Our observations support the notion that astrocyte microdomain Ca$^{2+}$ transients are regulated by extrinsic cues that can simultaneously stimulate two astrocytes, or that astrocyte-

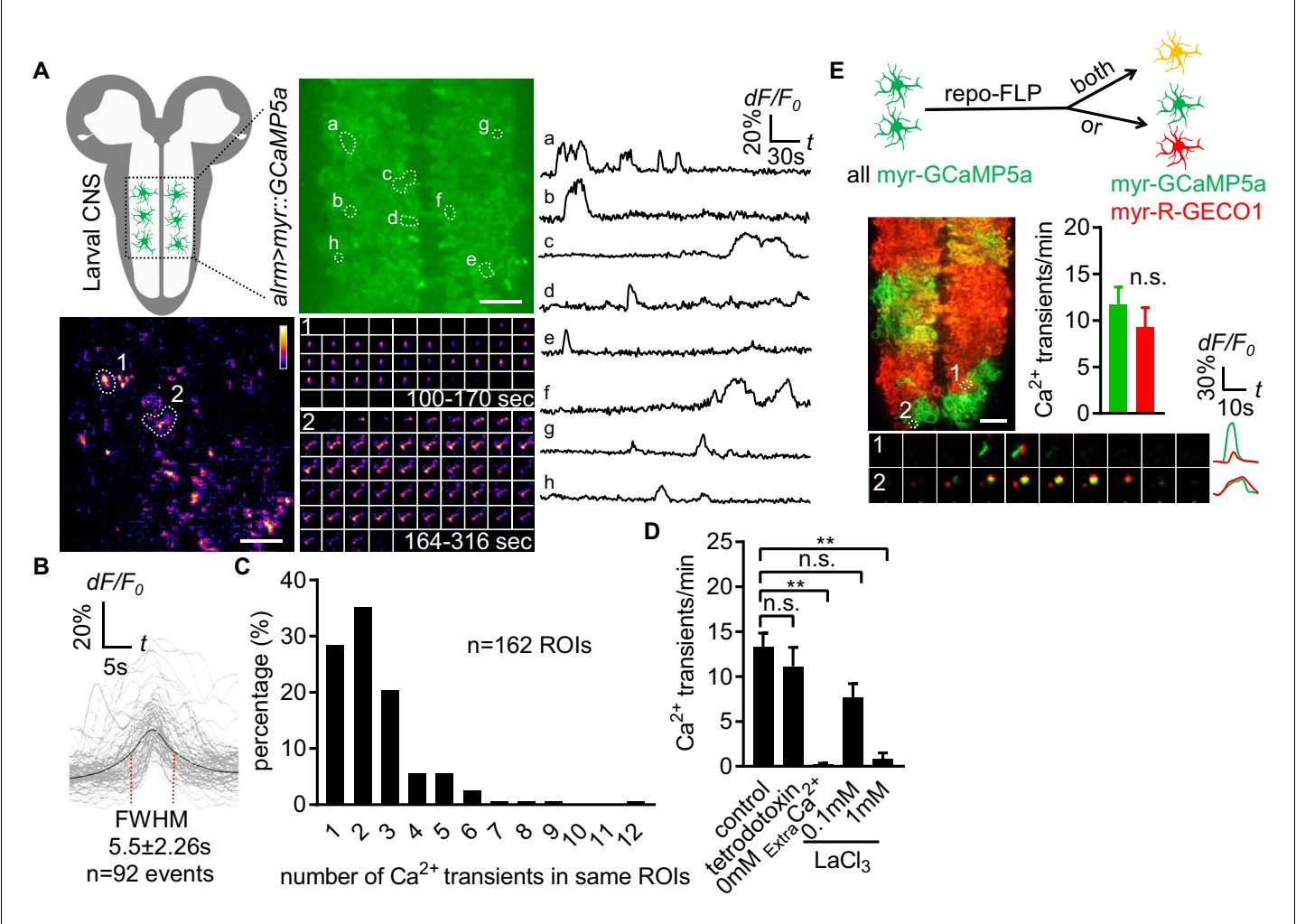

**Figure 1.** Characterization of microdomain Ca[2+] transients in *Drosophila* astrocytes. (**A**) Schematic of larval CNS (white area, neuropil; gray, cortex). An imaging area showing membrane tethered myr-GCaMP5a (green) in astrocytes, in which microdomain Ca[2+] transients during 6 min were maximally projected. Traces of 8 individual ROIs microdomains (right, **a–h**) are shown over the entire 6 min window. Pseudocolor grayscale of Ca[2+] signals (bottom left), grayscale values ranging from 0 to 255 (scale bars, 20 µm). Representative time-lapse images (bottom right) of two indicated microdomain Ca[2+] transients in 2 ROIs (1, 2). (**B**) Superimposed traces of individual microdomain Ca[2+] transients and an average with its full width at half maximum (FWHM, mean ± SD). (**C**) Histogram showing the distribution of recurrent microdomain Ca[2+] transients at same ROIs. (**D**) Responses of microdomain Ca[2+] transients to tetrodotoxin, 0 extracellular Ca[2+] and LaCl₃ (n = 6, mean ± SEM, one-way ANOVA). (**E**) 2-color Ca[2+] imaging in neighboring astrocytes. In presence of the flippase *repo-FLP*, myr-GCaMP5a expressing astrocytes switch to express myr-R-GECO1, resulting in 3 types of astrocytes: myr-GCaMP5a[+] (green), myr-R-GECO1[+] (red), expressing both indicators (orange) (scale bar, 20 µm). Quantification is from Ca[2+] imaging of myr-GCaMP5a and myr-R-GECO1 that were exclusively expressed in adjacent astrocytes (n = 3, unpaired t-test). Time-lapse images and superimposed traces of representative microdomain Ca[2+] transients at two juxtaposed ROIs between myr-GCaMP5a and myr-R-GECO1 expressing astrocytes. See source data *Figure 1—source data 1*.

The online version of this article includes the following video, source data, and figure supplement(s) for figure 1:

**Source data 1.** Characterization of microdomain Ca[2+] transients in *Drosophila* astrocytes.

**Figure supplement 1.** Spontaneous astrocyte microdomain Ca[2+] transients in larvae.

**Figure 1—video 1.** Microdomain Ca[2+] transients in acutely dissected CNS preparations from wildtype 3[rd] instar (L3) larva.

https://elifesciences.org/articles/58952#fig1video1

**Figure 1—video 2.** Microdomain Ca[2+] transients in intact CNS preparation from wildtype 1[st] instar larva.

https://elifesciences.org/articles/58952#fig1video2

astrocyte communication/coupling is sufficient to coordinate very local Ca$^{2+}$ signaling events across neighboring cells.

## Astrocyte microdomain Ca$^{2+}$ transients are enhanced by tyramine through TyrRII and mediated by TrpML

To determine whether neurotransmitters were capable of modulating astrocyte microdomain Ca$^{2+}$ transients, we bath applied several neurotransmitters and live-imaged astrocyte microdomain Ca$^{2+}$ events. Application of glutamate, acetylcholine, GABA, or octopamine had no effect on the frequency of astrocyte microdomain Ca$^{2+}$ transients (*Figure 2A*). In contrast, application of tyramine led to a significant increase in the frequency of these transients by ~40% (*Figure 2B*). We screened the known receptors for tyramine in *Drosophila* and found that astrocyte-specific depletion of TyrRII blocked the ability of tyramine to increase astrocyte microdomain Ca$^{2+}$ transients. The spontaneous microdomain events were not dependent on the presence of tyramine or octopamine, as mutants that block the production of tyramine and octopamine (*Tdc2$^{RO54}$*) or octopamine (*Tβh$^{nM18}$*) did not significantly alter the frequency of astrocyte microdomain Ca$^{2+}$ transients, nor did mutations in *Oct-TyrR*, which we previously showed was essential for activation of whole-cell Ca$^{2+}$ transients in astrocytes (*Figure 2C*). These data indicate that while astrocyte microdomain Ca$^{2+}$ transients can be partially enhanced by tyramine through TyrRII, under basal conditions astrocytes do not require tyramine or octopamine for microdomain Ca$^{2+}$ transient activity.

Whole-cell astrocyte transients are regulated by the TRP channel Water witch (Wtrw) (*Ma et al., 2016*), and astrocyte basal Ca$^{2+}$ levels in mammals are modulated by TrpA1 (*Shigetomi et al., 2013*). The molecular pathways that generate astrocyte microdomain Ca$^{2+}$ transients have not been identified. We speculated that astrocyte microdomain Ca$^{2+}$ transients might be regulated by one or more of the 13 TRP channels encoded in the *Drosophila* genome. We screened these for potential roles in the regulation of astrocyte microdomain Ca$^{2+}$ transients in animals bearing TRP channel mutations or astrocyte-specific RNAi targeting *trp* family genes. While knockout of 11 of these TRP channels had no effect, we found that microdomain Ca$^{2+}$ events decreased by ~70% to 80% in *trpml* loss-of-function mutants, in both intact 1$^{st}$ instar larvae (*Figure 2—figure supplement 1 - Figure 2A*) and acute CNS preparations from 3$^{rd}$ instar larvae (*Figure 2D*; *Figure 2—video 1*). Astrocyte-specific knockdown of *trpml* also reduced microdomain Ca$^{2+}$ events by ~60% (*Figure 2D*), and expressing a version of myc-tagged TrpML (*trpml-myc*) in astrocytes rescued decreased microdomains in *trpml* mutants (*Figure 2E*; *Figure 2—figure supplement 1 - Figure 2D*), arguing for a cell-autonomous role of TrpML in regulating astrocyte microdomain Ca$^{2+}$ transients. Although application of tyramine still increased microdomain Ca$^{2+}$ events in *trpml$^1$* mutants, the enhancement above the basal level of spontaneous microdomain Ca$^{2+}$ transients was significantly reduced (*Figure 2F*), suggesting tyramine enhances microdomains in frequency at least in part via TrpML.

We next examined whether TrpML was essential for tyramine activated Ca$^{2+}$ events in astrocyte cell bodies, and found that tyramine induced comparable Ca$^{2+}$ rise in astrocytes in control and *trpml$^1$* mutants (*Figure 2—figure supplement 1 – Figure 2C*). This argues for a specific role of TrpML in generating microdomain Ca$^{2+}$ transients. While glial cells undergo apoptosis in *trpml* mutant adults (*Venkatachalam et al., 2008*), astrocyte development appears to be normal at 3$^{rd}$ instar larval stage (*Figure 2—figure supplement 1 - Figure 2D*). Interestingly, *trpml$^1$* mutants exhibit near fully penetrant lethality during later pupal stages as previously reported (*Venkatachalam et al., 2008*), and we found that expression of *trpml-myc* selectively in astrocytes rescued this lethality (*Figure 2—figure supplement 1 - Figure 2E*), demonstrating that TrpML plays an essential role in astrocytes. The myc-tagged version of TrpML predominantly localized at GFP-Lamp1$^+$ endo-lysosomes in astrocytes in both fine astrocytic processes and cell bodies, as well as near the cell membrane (*Figure 2—figure supplement 1 – Figure 2F*). TrpML might therefore execute its regulation of astrocyte microdomains from endo-lysosomes, the plasma membrane, or both.

Previous work has shown that reactive oxygen species (ROS) can activate TrpML (*Zhang et al., 2016*), and astrocyte near-membrane Ca$^{2+}$ events in mammals are sensitive to ROS (*Agarwal et al., 2017*). We therefore assayed the sensitivity of *Drosophila* astrocyte microdomain Ca$^{2+}$ transients to ROS. We observed that bath application of the ROS generator hydrogen peroxide (H$_2$O$_2$) led to a TrpML-dependent increase in astrocyte microdomain Ca$^{2+}$ events, while, reciprocally, addition of the antioxidant N-acetyl cysteine completely abolished them (*Figure 2G and H*; *Figure 2—videos 2 and 3*). These data indicate that astrocyte microdomain Ca$^{2+}$ transients are mediated by TrpML and

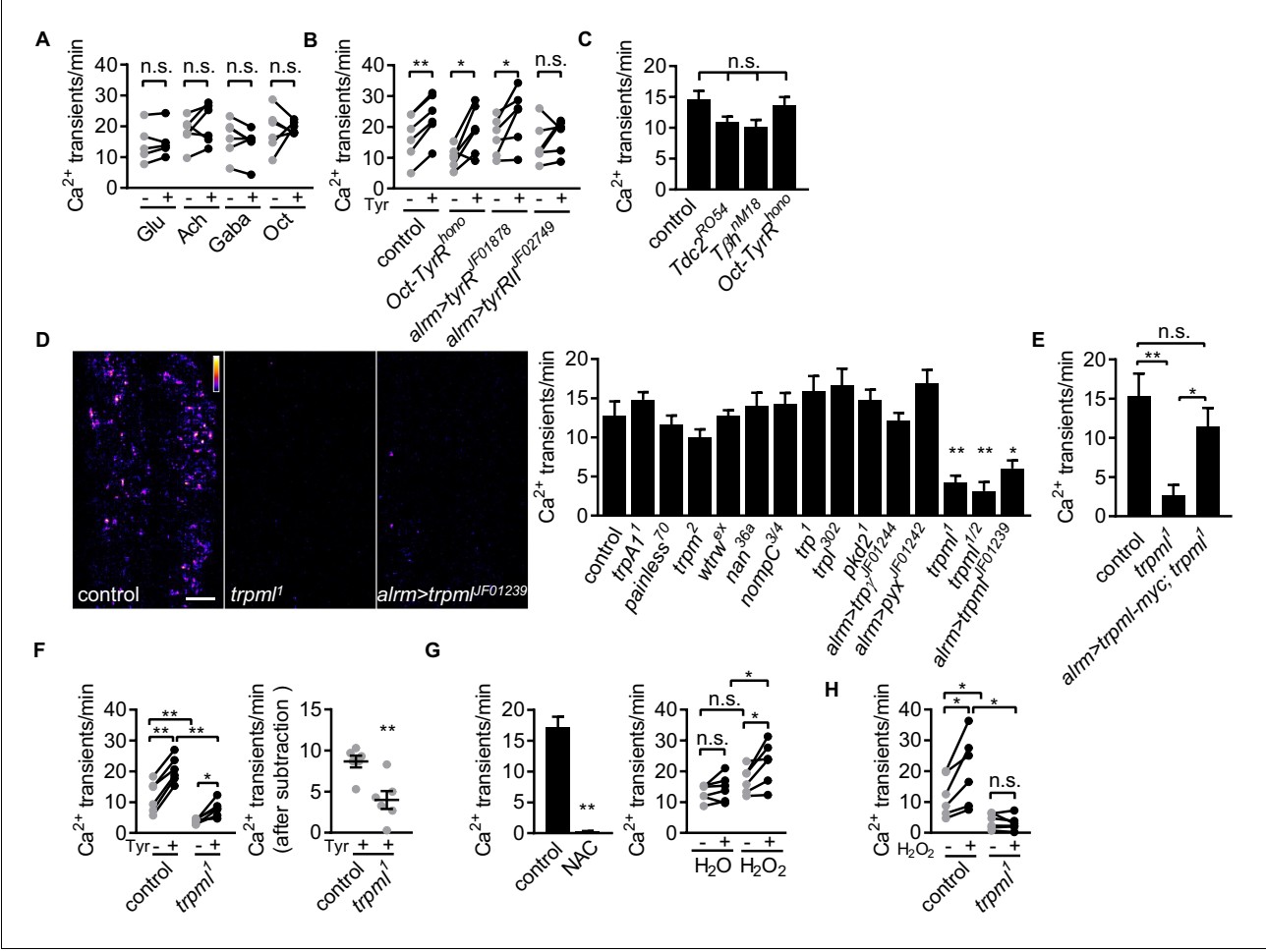

**Figure 2.** Astrocyte microdomain Ca²⁺ transients are genetically distinct from soma transients and require TrpML. (**A**) Responses of microdomain Ca²⁺ transients (frequency) to glutamate (Glu), acetylcholine (Ach), γ-aminobutyric acid (Gaba), tyramine (Tyr), octopamine (Oct) in presence of tetrodotoxin. (**B**) Effect of tyramine on Ca²⁺ transient frequency in genotypes indicated. *Oct-TyrR* mutants, and astrocyte-specific (*alrm>*) expression of RNAis to TyrR or TyrRII (in A and B, n = 6, mean ± SEM, paired t-test). (**C**) Quantification of microdomain Ca²⁺ transients in *Tdc2^RO54^*, *Tβh^nM18^* or *Oct-TyrR^hono^* mutants (n = 6, mean ± SEM, one-way ANOVA). (**D**) Maximally projected astrocyte microdomain Ca²⁺ transients during 6 min in control, *trpml¹* mutants, and astrocyte-specific *trpml^RNAi^* (in pseudocolor, grayscale values ranging from 0 to 255. scale bar, 20 μm). Quantification of microdomain Ca²⁺ transients in mutants (loss-of-function mutations or astrocyte-specific RNAi driven by *alrm-Gal4*) of genes encoding TRP family ion channels (n = 6, mean ± SEM, one-way ANOVA). (**E**) *trpml-myc* expression in astrocytes rescues microdomain Ca²⁺ transients in *trpml¹* mutants (n = 6, mean ± SEM, one-way ANOVA). (**F**) Effect of tyramine treatment on controls and *trpml¹* mutants (n = 6, mean ± SEM, one-way ANOVA. within groups, paired t-test). Right panel, increase in transients by tyramine (subtracting the basal included) in control and *trpml¹* mutants. (**G**) Application of the antioxidant N-acetyl cysteine (NAC) or H₂O₂ to the larval CNS. (for F and G n = 6, mean ± SEM, across groups, one-way ANOVA; within groups, paired t-test; -, + indicate pre-, post-delivery). (**H**) H₂O₂ application to control and *trpml* mutants shows that H₂O₂-dependent increases require TrpML. See source data

*Figure 2—source data 1*.

The online version of this article includes the following video, source data, and figure supplement(s) for figure 2:

**Source data 1.** Astrocyte microdomain Ca²⁺ transients require TrpML.

**Source data 2.** Astrocyte microdomain Ca²⁺ transients are genetically distinct from soma transients and require TrpML.

**Figure supplement 1.** Astrocyte microdomain Ca²⁺ transients require TrpML.

**Figure 2—video 1.** Microdomain Ca²⁺ transients in acutely dissected CNS preparations from *trpml* mutant 3ʳᵈ instar (L3) larva.

https://elifesciences.org/articles/58952#fig2video1

**Figure 2—video 2.** Microdomain Ca²⁺ transients pre/post H₂O₂ treatment in wild type 3ʳᵈ instar (L3) larva.

https://elifesciences.org/articles/58952#fig2video2

**Figure 2—video 3.** Microdomain Ca²⁺ transients pre/post H₂O₂ treatment in *trpml¹* mutant 3ʳᵈ instar (L3) larva.

https://elifesciences.org/articles/58952#fig2video3

are highly sensitive to ROS. Furthermore, our observation that TrpML and Wtrw regulate only micro-domain or whole-cell $Ca^{2+}$ events, respectively, demonstrates that astrocyte microdomain $Ca^{2+}$ transients and whole-cell changes in astrocyte $Ca^{2+}$ are physiologically and genetically distinct signaling events, although adrenergic transmitters (tyramine in *Drosophila* and norepinephrine in mouse) may serve as factors to coordinate their activity.

## Astrocyte microdomain $Ca^{2+}$ transients are associated with trachea and precede tracheal filopodia retraction

Mammalian astrocytes make intimate contacts with blood vessels by forming endfeet to allow for gas exchange, uptake of nutrients from blood, and maintenance of the blood brain barrier. Fine astrocyte processes in *Drosophila* infiltrate the CNS neuropil where they associate with neural processes, synapses, and tracheal elements (*Freeman, 2015*). Tracheal cells serve a similar function to mammalian blood vessels, and their development and morphogenesis are molecularly similar (*Ghabrial et al., 2003*). Trachea are an interconnected series of gas-filled tubes that penetrate insect tissues, and gas exchange occurs through tracheal cell–tissue interactions (*Ghabrial et al., 2003*). Interestingly, we observed that half of all astrocyte microdomain $Ca^{2+}$ transients we recorded were closely associated with CNS tracheal elements (*Figure 3A*). In live preparations where trachea were labeled with myristoylated tdTomato (myr-tdTom) and either Lifeact-GFP to visualize actin or Tubu-lin-GFP to visualize microtubules, we observed that tracheal branches dynamically extended and retracted actin-rich protrusions that are characteristic of filopodia (*Figure 3B*), and only very few were stabilized by microtubules (*Figure 3—figure supplement 1 - Figure 3A*). These observations imply tracheal branches dynamically explore their surroundings in the CNS with filopodia.

We classified tracheal filopodia into four categories according to different behavior they exhibited during imaging: extension (1.7%), retraction (20.7%), extension and retraction (75.3%), or stationary (2.3%) (*Figure 3C*). The vast majority of tracheal filopodia dynamically extended and retracted during the imaging window of 6 min. We noted that those that exhibited only retraction did so very early in the imaging window, which could indicate that we began our imaging after extension had been initiated, however we cannot exclude the possibility that our imaging approach biases tracheal dynamics more toward retraction (e.g. by our imaging procedure generating ROS).

Based on their close association, we explored the potential relationship between tracheal filopo-dial dynamics and astrocyte microdomain $Ca^{2+}$ transients. Interestingly, 52% of tracheal filopodial tips overlapped, at some point, with an astrocyte microdomain $Ca^{2+}$ transient (*Figure 3C*). Overlap was defined as the tracheal filopodial tip falling within the maximum size of the domain of the astro-cyte $Ca^{2+}$ transient. Moreover, we observed that astrocyte microdomain $Ca^{2+}$ transients preceded retraction events of tracheal filopodia within their domain. The onset of trachea filopodial retraction was tightly correlated with astrocyte microdomain $Ca^{2+}$ peaks ($R^2 = 0.99$, *Figure 3D–F*; *Figure 3—figure supplement 1 – Figure 3B*; *Figure 3—video 1*), with a latency time of $25.9 \pm 2.18$ s. In contrast, the intervals between trachea filopodial extension onset and astrocyte microdomain $Ca^{2+}$ peaks were significantly larger ($144.6 \pm 11.65$ s) and more broadly distributed (*Figure 3—figure supplement 1 - Figure 3C*). Correlations between astrocyte microdomains and any changes in tracheal filopodial dynamics were only observed when filopodia overlapped with astrocyte microdomain $Ca^{2+}$ transients. For instance, we found no correlation between extension or retraction of filopodia and nearby non-overlapping astrocyte microdomain $Ca^{2+}$ transients ('bystanders') within an circular area beginning 7.5 μm away from filapodial tips and extending outward (*Figure 3—figure supplement 1 - Figure 3D*). We noted that 46% of tracheal filopodia were not visibly associated with astro-cyte microdomain $Ca^{2+}$ transients. This argues that a large fraction of tracheal filopodia can extend and retract in the absence of local astrocyte microdomain $Ca^{2+}$ signaling. However, we cannot exclude the possibility that astrocyte $Ca^{2+}$ signaling above or below the plane of focus could be modulating the dynamics of these trachea. Together, these observations indicate that a large fraction of astrocyte microdomain $Ca^{2+}$ transients are spatiotemporally correlated with the retraction of adjacent tracheal filopodia.

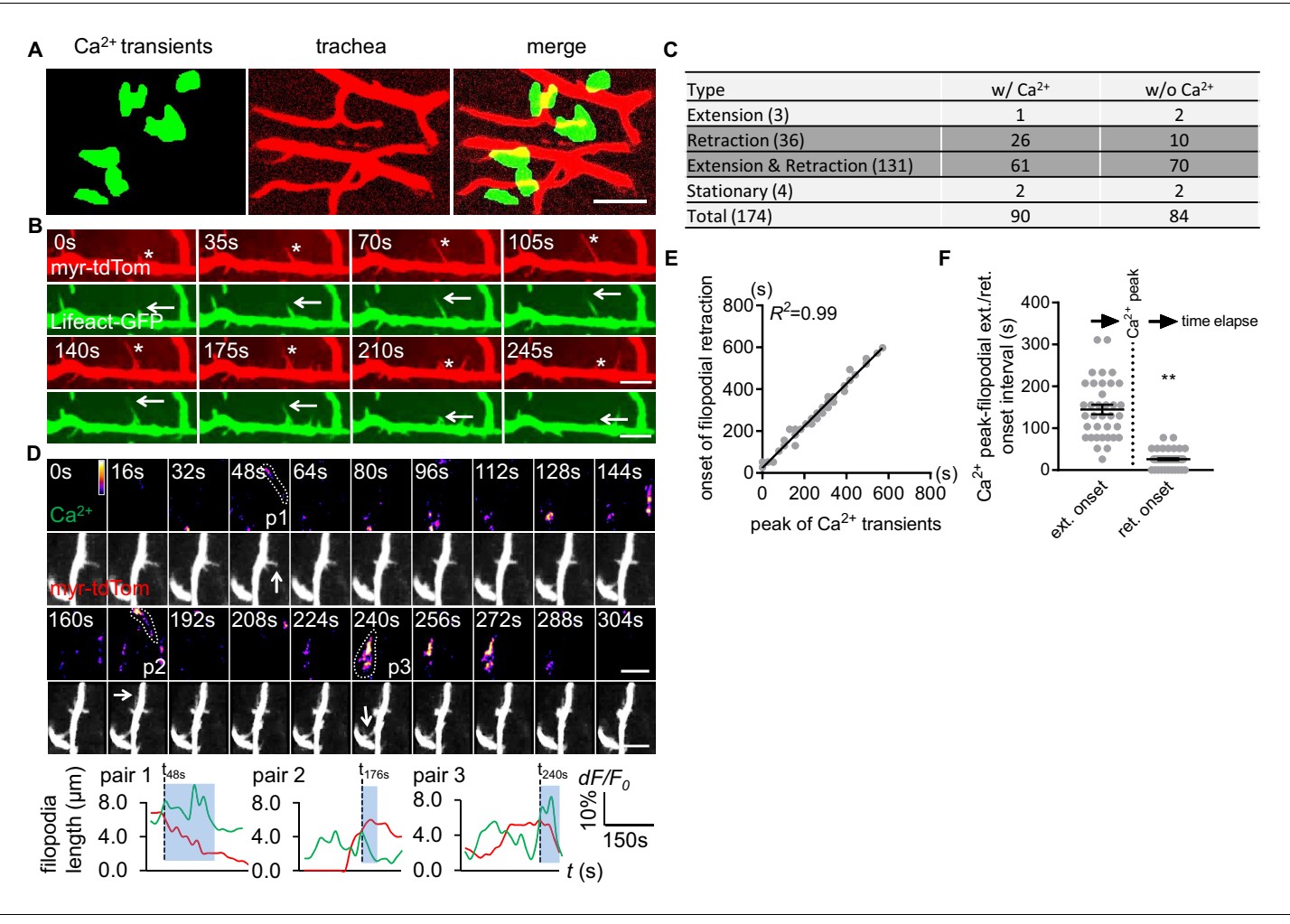

**Figure 3.** Astrocyte microdomain Ca²⁺ transients are associated with tracheal branches and precede retraction onset of tracheal filopodia. (A) Astrocyte microdomain Ca²⁺ transients (green) overlap with tracheal branches (red) (scale bar, 20 µm). (B) Tracheal branches extend and retract F-actin containing filopodia. Asterisks, myr-tdTomato labeled filopodia; Arrows, Lifeact-GFP labeled F-actin (scale bar, 10 µm). (C) Categorization of tracheal filopodia across the entire population (extension, retraction, extension and retraction, or stationary). w/Ca²⁺ indicates tracheal filopodia extended into an astrocyte microdomain Ca²⁺ transient; w/o Ca²⁺ indicates no visible astrocyte microdomain Ca²⁺ transient was observed. (D) Time-lapse images and superimposed traces (green trace, myr-GCaMP5a in astrocytes expressed as $dF/F_0$; red trace, myr-tdTomato in tracheal filopodia expressed as length) of 3 pairs (p1, p2, p3) of tracheal filopodia and astrocyte microdomain Ca²⁺ transients (in pseudocolor, grayscale values ranging from 0 to 255. scale bar, 10 µm). Vertical dash lines, timepoints when tracheal filopodia enter astrocyte Ca²⁺ microdomains. Blue boxes, time windows astrocyte microdomain Ca²⁺ transients persist after tracheal filopodia enter. Note that prior to entering the astrocyte Ca²⁺ microdomain, tracheal filopodial extension is not coupled to increases in astrocyte Ca²⁺, but after entry, increased astrocyte Ca²⁺ is strongly correlated with tracheal filopodial retraction. (E) Temporal correlation between onset of filopodia retraction and timing of peak astrocyte microdomain Ca²⁺ transients in seconds (n = 61 filopodia from Extension and Retraction pool). (F) Time intervals between astrocyte microdomain Ca²⁺ transients and tracheal filopodial extension versus retraction. See source data *Figure 3—source data 1*.

The online version of this article includes the following video, source data, and figure supplement(s) for figure 3:

**Source data 1.** Distribution of microtubules and F-actin in larval CNS tracheal elements.

**Source data 2.** Astrocyte microdomain Ca²⁺ transients are associated with tracheal branches and precede retraction onset of tracheal filopodia. hea.

**Figure supplement 1.** Distribution of microtubules and F-actin in larval CNS trachea.

**Figure 3—video 1.** Microdomain Ca²⁺ transients and tracheal filopodial dynamics in wildtype L3 CNS preparation, related to *Figure 3D*.

https://elifesciences.org/articles/58952#fig3video1

# Blockade of astrocyte microdomain Ca²⁺ transients increases CNS ROS and *trpml* mutants exhibit increased tracheal growth

Based on their spatiotemporal association, we speculated that astrocyte microdomain Ca$^{2+}$ transients promote tracheal filopodial retraction in response to ROS through TrpML. This predicts that loss of these transients would increase tracheal filopodial growth. To block astrocyte microdomain Ca$^{2+}$ transients we used *trpml$^1$* mutants and labeled tracheal membranes with myr-tdTom. We found in *trpml$^1$* mutants that the overall rate of filopodial extension over time was indeed increased, which resulted in an increase in maximum length of the tracheal filopodia (***Figure 4A***). The increased filopodial extension rate, but not the maximum length of the filopodia, was phenocopied by knocking down *trpml* selectively in either astrocytes, and to some extent in trachea (***Figure 4—figure supplement 1 – Figure 4A***), arguing that TrpML functions in both astrocytes and trachea to control tracheal filopodial growth. Filopodial retraction rates remained unchanged in *trpml$^1$* mutants, suggesting astrocyte TrpML signaling facilitates tracheal filopodial retraction by suppressing extension. Our model further predicts that stimulating an increased number of astrocyte microdomain Ca$^{2+}$ transients should promote filopodial retraction. To test this idea we bath applied tyramine, which stimulates astrocyte microdomain Ca$^{2+}$ transients. We found tyramine application led to an increase in the percentage of tracheal filopodial retracting versus extending (***Figure 4B***), further supporting the notion that astrocyte microdomains facilitate filopodial retraction.

To quantify the longer term structural effect of loss of *trpml* (and astrocyte microdomain Ca$^{2+}$ transients), we examined tracheal structure in larval CNS. We first counted the total number of protrusions from a pair of most posterior ganglion trachea (mpgTr) that innervate a few segments from A5 to A8/9 in the ventral nerve cord. We found that mpgTr in the ventral nerve cord in *trpml$^1$* mutants exhibited increased total length compared to those in control animals (***Figure 4—figure supplement 1 - Figure 4B***). We next examined a uniquely identifiable branch of the tracheal system in the larval optic neuropil (LON) (***Sprecher et al., 2011***). The LON is a simple tissue, composed of only a few dozen neurons and 1 ~ 2 tracheal branches that are surrounded by the processes from a single astrocyte (***Figure 4—figure supplement 1 - Figure 4C***). Compared to controls, we found that *trpml$^1$* mutants exhibited an approximate doubling of the number of tracheal branches, and also total filopodia in the LON (***Figure 4C***). Together these data indicate that TrpML restricts tracheal outgrowth through astrocyte microdomain Ca$^{2+}$ transients.

It is plausible that such an increase in tracheal branches and filopodia might result in excessive O$_2$ delivery to CNS tissues and result in a hyperoxia. To assay ROS status in the CNS we stained with the ROS indicator dihydroethidium (DHE). In live preparations of *trpml$^1$* mutants we found a ~ 3 fold increase in oxidized DHE$^+$ puncta, likely in mitochondria (***Bindokas et al., 1996***), and when we eliminated *trpml* selectively from astrocytes by RNAi, we found a ~ 2.5 fold increase in oxidized DHE$^+$ puncta (***Figure 4D***). We also validated our DHE staining by a 5 min period of acute pre-treatment with the ROS scavenger NAC, which reduced DHE$^+$ puncta in *trpml$^1$* mutants (***Figure 4D***; ***Figure 4—figure supplement 1 – Figure 4D***). These data support the notion that blockade of TrpML signaling, and in turn astrocyte microdomain Ca$^{2+}$ transients, leads to increased ROS in the CNS, perhaps due to excessive O$_2$ delivery from increased tracheal elements (***Figure 4E***).

## Discussion

Molecules required for the generation of astrocyte microdomain Ca$^{2+}$ transients have remained elusive, it remains unclear how many 'types' of microdomain Ca$^{2+}$transients exist in astrocytes, and the in vivo roles for these transients remain controversial and poorly defined (***Agarwal et al., 2017***; ***Bazargani and Attwell, 2016***). Our work demonstrates that *Drosophila* astrocyte microdomain Ca$^{2+}$ transients are mediated by the TRP channel TrpML, and can be stimulated by ROS and tyramine through the TyrRII receptor. Unexpectedly, we found that a large fraction of astrocyte microdomain Ca$^{2+}$ transients are closely associated with tracheal elements, precede and are tightly coupled with tracheal filopodial retraction, and that stimulating astrocyte microdomain Ca$^{2+}$ transients with tyramine can promote tracheal filopodial retraction. We propose that one important physiological role for a subset of astrocyte microdomain Ca$^{2+}$ transients is to modulate CNS gas exchange through TrpML and ROS signaling.

Astrocyte microdomain Ca$^{2+}$ transients in *Drosophila* share many features with those observed in mammals (***Agarwal et al., 2017***; ***Nett et al., 2002***; ***Srinivasan et al., 2015***; ***Zhang et al., 2017***).

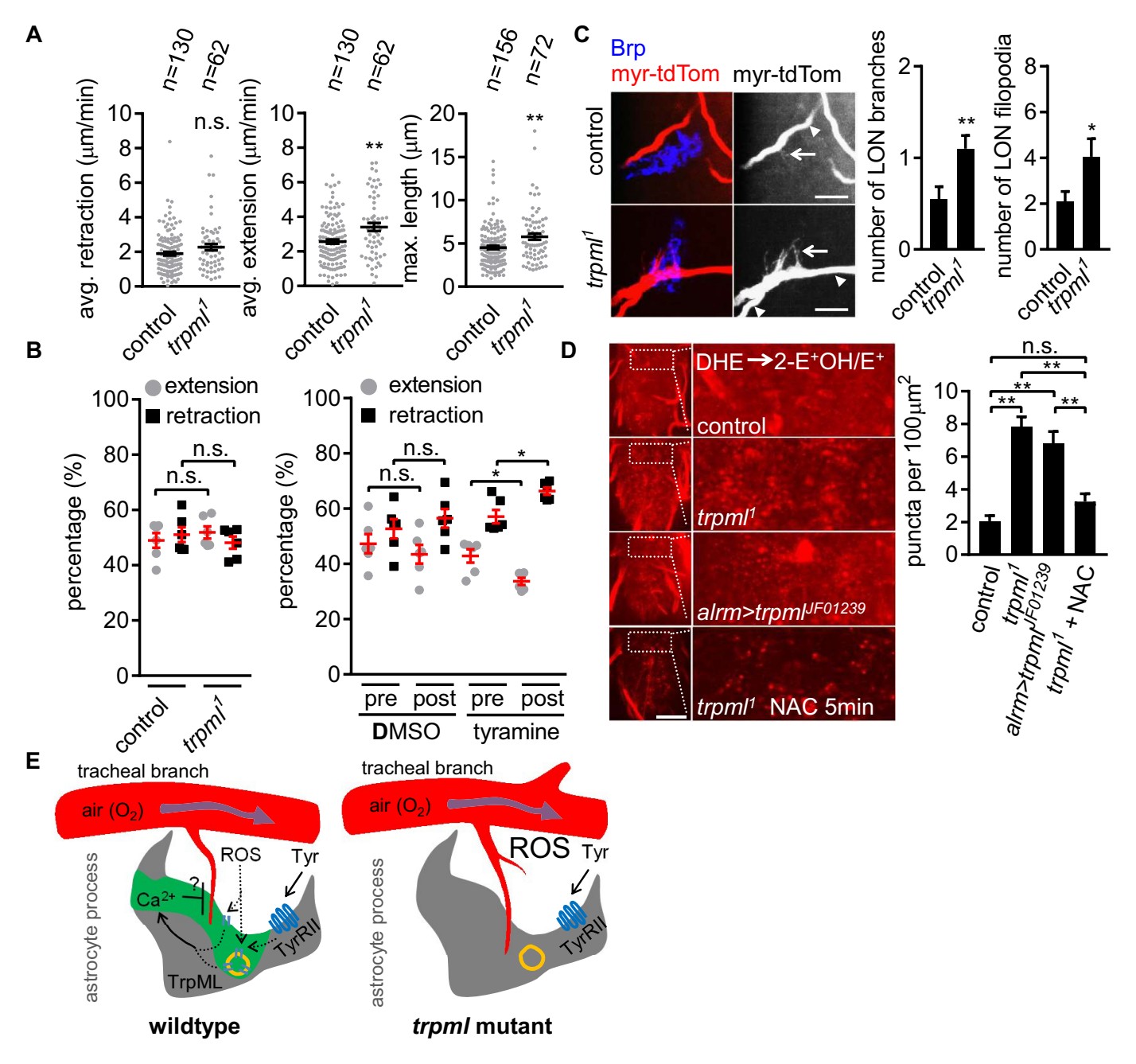

**Figure 4.** Loss of TrpML leads to overgrowth of trachea and excessive reactive oxygen species (ROS) in larval CNS. (**A**) Quantification of average filopodial retraction and extension rates, and maximal length of tracheal filopodia (n indicate the number of filopodia randomly selected from 6 to 8 larval CNS, mean ± SEM, t-test). (**B**) Comparison of changes in extension/retraction ratios after bath application of tyramine in control and *trpml¹* mutants (n = 6, mean ± SEM, one-way ANOVA across groups; within groups, paired t-test). (**C**) In control, one tracheal branch (labeled with *btl >myr:: tdTom*, arrowhead) near the LON (Brp, blue) grows short filopodia (arrows) into the LON. In *trpml¹*, two transverse tracheal branches grow near the LON, and they exhibit increases in filopodial extension into the LON (n = 20, mean ± SEM, t-test. scale bar, 10 μm). (**D**) Dihydroethidium (DHE) staining in indicated genotypes. ROS-oxidized DHE forms 2-hydroxyethidium (2-OH-E⁺) and ethidium (E⁺). Quantifications to right (n = 6, mean ± SEM, t-test. scale bar, 50 μm). NAC was added 5 min prior to DHE incubation. (**E**) Proposed model. Astrocyte microdomain $Ca^{2+}$ transients are modulated by reactive oxygen species (ROS) and TrpML. Astrocyte $Ca^{2+}$ signaling can facilitate filopodial retraction if a tracheal filopodium enters a microdomain. Cell cortex underlying membrane or lysosomal localization of TrpML regulates $Ca^{2+}$ signaling in response to ROS, presumably due to increased $O_2$ delivery, to generate astrocyte microdomain $Ca^{2+}$ transients. Tyramine induces increased microdomain $Ca^{2+}$ transients via TrpML. Loss of astrocyte microdomains resulted from *trpml* mutation leads to overgrown trachea including filopodia, and increased ROS. See source data *Figure 4—source data 1*.

*Figure 4 continued on next page*

*Figure 4 continued*

The online version of this article includes the following source data and figure supplement(s) for figure 4:

**Source data 1.** Loss of TrpML leads to overgrowth of trachea and excessive reactive oxygen species (ROS) in larval CNS.
**Source data 2.** Tracheal branches overgrow in the larval ventral nerve cord of *trpml¹* mutants.
**Figure supplement 1.** Tracheal branches overgrow in the larval ventral nerve cord of *trpml¹* mutants.

They are spontaneously generated, exhibit diverse waveforms, and appear for the most part asynchronous across astrocyte populations. Their production requires the presence of extracellular $Ca^{2+}$ and they are suppressed by the broad $Ca^{2+}$ channel blocker $LaCl_3$. Our analysis of adjacent astrocytes that are labeled with uniquely identifiable $Ca^{2+}$ indicators (myr-GCaMP5a versus myr-R-GECO1) demonstrated individual astrocyte microdomain $Ca^{2+}$ transients can span astrocyte-astrocyte boundaries, which might argue for an extrinsic mechanism regulating their production. Alternatively, it might indicate that astrocyte-astrocyte coupling of $Ca^{2+}$ signaling events is strong during their production. Astrocyte microdomain $Ca^{2+}$ transients are not suppressed by blockade of action potentials with tetrodotoxin, suggesting they are not activity dependent, although we also cannot formally rule out a role for spontaneous release of neurotransmitters at synapses.

Based on their persistence in *Tdc2* mutants, which lack tyramine and octopamine in CNS, we conclude that spontaneous astrocyte microdomain $Ca^{2+}$ transients do not require the production of tyramine or octopamine in vivo. Nevertheless, and similar to application of norepinephrine to mouse preparations (*Agarwal et al., 2017*), we found that tyramine was capable of stimulating a partial increase in astrocyte microdomain $Ca^{2+}$ transients, and that to some extent this required the tyramine receptor TyrRII on astrocytes. Why these transients do not require tyramine for their spontaneous production, but can be stimulated by tyramine application, remains unclear, but this adrenergic regulation of astrocyte microdomain $Ca^{2+}$ signaling appears to be a conserved feature in *Drosophila* and mouse. It is possible that adrenergic regulation may serve as a way to physiologically couple whole-cell and microdomain $Ca^{2+}$ transients, as a mechanism to coordinate neuronal activity with astrocyte calcium signaling, or both. The regulation of astrocyte $Ca^{2+}$ signaling in vivo by octopamine and tyramine is complex. Tyramine can stimulate increases in whole-cell $Ca^{2+}$ levels in astrocytes through the Oct-TyrR, as does octopamine (*Ma et al., 2016*), but the latter has no effect in astrocyte microdomain $Ca^{2+}$ transients. Signaling mediated by TyrRII may somehow increase the open probability of TrpML in microdomains in a way that is distinct from whole-cell $Ca^{2+}$ signaling. This is consistent with our observation that TrpML is partially responsible for the tyramine-induced increase in astrocyte microdomain $Ca^{2+}$ transients.

TrpML may exert its effects on astrocytic microdomain $Ca^{2+}$ transients at endo-lysosomes, at the cell cortex, or both (*Figure 4E*). TrpML is well known to signal in the endo-lysosomal compartment as well as the plasma membrane (*Feng et al., 2014*). We observed localization of TrpML to endo-lysosomes and the cell surface (or cell cortex) in *Drosophila* astrocytes. TrpML may function in astrocytic endo-lysosomes (e.g. $Ca^{2+}$ release from these compartments) to indirectly activate $Ca^{2+}$ entry through other cation channels at the cell surface. Alternatively, TrpML might function at the cell surface to directly drive $Ca^{2+}$ entry, or stimulate entry from the extracellular space by other channels, or release from intracellular stores. It is also important to note that while we have assayed $Ca^{2+}$ entry, TrpML is also capable of passing several other cations (*Feng et al., 2014*), and so its role in regulating astrocyte functions could be mediated by these cations in addition to $Ca^{2+}$.

Astrocyte microdomain $Ca^{2+}$ transients and whole-cell changes in astrocyte $Ca^{2+}$ signaling appear to be distinct, in terms of their regulation by neurotransmitters and the molecular machinery generating each type of event (*Agarwal et al., 2017*; *Bazargani and Attwell, 2016*; *Srinivasan et al., 2015*). In mammals, whole-cell astrocyte $Ca^{2+}$ transients are modulated by norepinephrine, adrenergic receptor signaling, and startle stimuli, while microdomain $Ca^{2+}$ transients are activity-independent, associated with mitochondria and are sensitive to ROS (*Agarwal et al., 2017*; *Ding et al., 2013*; *Paukert et al., 2014*). Similarly, whole-cell transients in *Drosophila* astrocytes are activated by tyramine or octopamine (the invertebrate adrenergic neurotransmitters). We previously showed either of these can activate the Oct-TyrR receptor and the TRP channel Waterwitch on astrocytes, and in turn through the adenosine receptor AdoR, mediate many of the physiological and behavioral changes exerted by tyramine and octopamine (*Ma et al., 2016*). In contrast, microdomain $Ca^{2+}$

transients in astrocytes are not regulated directly by neural activity, are mediated by TrpML (but not Oct-TyrR or Wtrw) and they are sensitive to ROS.

In mouse, a subset of microdomain $Ca^{2+}$ transients are associated with mitochondria, which have been proposed to serve as a source of ROS, potentially through transient opening of the mPTP (*Agarwal et al., 2017*). These events may also require mitochondria in *Drosophila*, however the density of mitochondria in astrocyte processes was sufficiently high in our preparations that drawing such a conclusion was not feasible—a single astrocyte microdomain $Ca^{2+}$ transient appears to span domains that include many mitochondria. It is reasonable to speculate that astrocyte microdomain $Ca^{2+}$ transients across species are to a significant extent functionally distinct from whole-cell fluctuations, and that they play a role in coupling astrocyte signaling with tracheal/vasculature dynamics or other metabolic changes occurring in astrocytic mitochondria (*Agarwal et al., 2017*). Multiple distinct functional roles for astrocyte microdomain $Ca^{2+}$ signaling (i.e. in tracheal regulation versus metabolism) could explain why although only half of all observed astrocyte microdomain $Ca^{2+}$ transients are associated with tracheal filopodia, all appear to be potently regulated by ROS. Perhaps the remaining half of astrocyte microdomain $Ca^{2+}$ transients not associated with trachea are responding to mitochondria-based metabolic needs through ROS signaling. Finally, approximately half of tracheal filopodia we observed were not visibly associated with astrocyte microdomain $Ca^{2+}$ transients, yet they were able to extend and retract. This argues that other mechanisms exist to modulate tracheal dynamics in the CNS and astrocyte microdomain $Ca^{2+}$ transients only represent one regulatory mechanism.

Maintaining a healthy, spatiotemporally regulated normoxic environment to prevent either hypoxia or hyperoxia in CNS is an enormous and ongoing challenge, as the $O_2$-consuming metabolism is thought to fluctuate vigorously in response to neural activity. The close association of astrocyte microdomain $Ca^{2+}$ transients with trachea, the larval breathing apparatus, and with tracheal retraction in particular, suggests a role in modulating CNS gas exchange. Increase in $O_2$ delivery to tissues can lead to hyperoxia and elevated production of ROS. TrpML has recently been found to be a ROS sensor (*Zhang et al., 2016*), which would allow for a simple mechanism for ROS-mediated activation of TrpML downstream of increased $O_2$ delivery. Consistent with such a role in regulation of gas exchange, we found an increase in ROS in the CNS of *trpml* mutants, and we demonstrated that bath application of tyramine to stimulate astrocyte microdomain $Ca^{2+}$ transients was sufficient to promote tracheal filopodial retraction. Together our data support a model (*Figure 4E*) where a subset of TrpML-mediated microdomain $Ca^{2+}$ transients in astrocytes facilitate tracheal retraction, likely in conjunction with TrpML in trachea. Complete loss of TrpML led to an increase in tracheal growth (e.g. an increased vascularization of neural tissues), which argues that astrocyte-tracheal signaling through TrpML can regulate long-term structural changes in the tracheal morphology. Based on these findings, we propose that an important role for a subset of astrocyte microdomain $Ca^{2+}$ transients in the larval CNS is to coordinate gas exchange through regulation of tracheal dynamics, thereby balancing $O_2$ delivery/$CO_2$ removal according to local metabolic needs.

## Materials and methods

**Key resources table**

| Reagent type (species) or resource | Designation | Source or reference | Identifiers | Additional information |
|---|---|---|---|---|
| Genetic reagent (*D. melanogaster*) | *trpml¹* | Bloomington Stock Center | BDSC: 28992 | FlyBase symbol: *w¹¹¹⁸; Trpml¹* |
| Genetic reagent (*D. melanogaster*) | *trpml²* | Bloomington Stock Center | BDSC: 42230 | FlyBase symbol: *w\*; Trpml²/TM6B, Tb¹* |
| Genetic reagent (*D. melanogaster*) | *trpml^JF01239* | Bloomington Stock Center | BDSC: 31294 | FlyBase symbol: *y¹ v¹; P{TRiP.JF01239}attP2* |
| Genetic reagent (*D. melanogaster*) | *tyrR^JF01878* | Bloomington Stock Center | BDSC: 25857 | FlyBase symbol: *y¹ v¹; P{TRiP.JF01878}attP2* |
| Genetic reagent (*D. melanogaster*) | *tyrRII^JF02749* | Bloomington Stock Center | BDSC: 27670 | FlyBase symbol: *y¹ v¹; P{TRiP.JF02749}attP2* |

*Continued on next page*

*Continued*

| Reagent type (species) or resource | Designation | Source or reference | Identifiers | Additional information |
|---|---|---|---|---|
| Genetic reagent (*D. melanogaster*) | *trpA1[1]* | Bloomington Stock Center | BDSC: 36342 | FlyBase symbol: *TI{TI}TrpA1[1]* |
| Genetic reagent (*D. melanogaster*) | *trpm[2]* | Bloomington Stock Center | BDSC: 35527 | FlyBase symbol: *w\*; TI{TI}Trpm[2]/CyO* |
| Genetic reagent (*D. melanogaster*) | *nompC[3]* | Bloomington Stock Center | BDSC: 42258 | FlyBase symbol: *nompC[3] cn[1] bw[1]/CyO* |
| Genetic reagent (*D. melanogaster*) | *trp[1]* | Bloomington Stock Center | BDSC: 5692 | FlyBase symbol: *trp[1]* |
| Genetic reagent (*D. melanogaster*) | *trpl[302]* | Bloomington Stock Center | BDSC: 31433 | FlyBase symbol: *cn[1] trpl[302] bw[1]* |
| Genetic reagent (*D. melanogaster*) | *pkd2[1]* | Bloomington Stock Center | BDSC: 24495 | FlyBase symbol: *w[1118]; Pkd2[1]/CyO* |
| Genetic reagent (*D. melanogaster*) | *trpγ[JF01244]* | Bloomington Stock Center | BDSC: 31299 | FlyBase symbol: *y[1] v[1]; P{TRiP.JF01244}attP2* |
| Genetic reagent (*D. melanogaster*) | *pyx[JF01242]* | Bloomington Stock Center | BDSC: 31297 | FlyBase symbol: *y[1] v[1]; P{TRiP.JF01242}attP2* |
| Genetic reagent (*D. melanogaster*) | *btl-Gal4* | Bloomington Stock Center | BDSC: 8807 | FlyBase symbol: *W\*; P{GAL4-btl.S}2, P{UASp-Act5C.T:GFP}2/ CyO, P{lacZ.w+}276. UASp-Act5C.T:GFP* was replaced with *10XUAS-IVS-myr::tdTomato* by recombination in this paper. |
| Genetic reagent (*D. melanogaster*) | *UAS-Lifeact-GFP* | Bloomington Stock Center | BDSC: 57326 | FlyBase symbol: *w\*; P{UAS-Lifeact.GFP.W}3* |
| Genetic reagent (*D. melanogaster*) | *UASp-αTub-GFP* | Bloomington Stock Center | BDSC: 7373 | FlyBase symbol: *w\*; P{UASp-GFPS65C-αTub84B}3/TM3, Sb[1]* |
| Genetic reagent (*D. melanogaster*) | *10XUAS-IVS-myr::tdTomato* | Bloomington Stock Center | BDSC: 32222 | FlyBase symbol: *w\*; P{10XUAS-IVS-myr::tdTomato}attP40* |
| Genetic reagent (*D. melanogaster*) | *UAS-trpml-myc* | Bloomington Stock Center | BDSC: 57372 | FlyBase symbol: *w\*; P{UAS-Trpml.MYC}3, Trpml[1]/TM6B, Tb[1]* |
| Genetic reagent (*D. melanogaster*) | *5XUAS-trpml-GCaMP5g* | Bloomington Stock Center | BDSC: 80066 | FlyBase symbol: *y[1] w\*; PBac{5XUAS-Trpml::GCaMP5G}VK00033* |
| Genetic reagent (*D. melanogaster*) | *UAS-GFP-Lamp1* | Bloomington Stock Center | BDSC: 42714 | FlyBase symbol: *w\*; P{UAS-GFP-LAMP}2; P{nSyb-GAL4.S}3/T(2;3)TSTL, CyO: TM6B, Tb[1]* |
| Genetic reagent (*D. melanogaster*) | *Oct-TyrR[hono]* | Kyoto Stock Center (DGRC) | BDSC: 109038 | FlyBase symbol: *w[1118]; P{lwB}Oct-TyrR[hono]* |
| Genetic reagent (*D. melanogaster*) | *btl-LexA* | *Roy et al., 2014* | | |
| Genetic reagent (*D. melanogaster*) | *nompC[4]* | *Walker et al., 2000* | | |
| Genetic reagent (*D. melanogaster*) | *painless[70]* | *Im et al., 2015* | | |
| Genetic reagent (*D. melanogaster*) | *nan[36a]* | *Kim et al., 2003* | | |
| Genetic reagent (*D. melanogaster*) | *wtrw[ex]* | *Kim et al., 2010* | | |
| Genetic reagent (*D. melanogaster*) | *Tdc2[RO54]* | *Cole et al., 2005* | | |
| Genetic reagent (*D. melanogaster*) | *Tβh[nM18]* | *Monastirioti et al., 1996* | | |

*Continued on next page*

*Continued*

| Reagent type (species) or resource | Designation | Source or reference | Identifiers | Additional information |
|---|---|---|---|---|
| Genetic reagent (D. melanogaster) | *alrm-Gal4* | *Doherty et al., 2009* | | |
| Genetic reagent (D. melanogaster) | *alrm > QF >* Gal4 | *Stork et al., 2014* | | |
| Genetic reagent (D. melanogaster) | *repo-FLPase* | *Stork et al., 2014* | | |
| Genetic reagent (D. melanogaster) | *alrm-LexA::GAD* | *Stork et al., 2014* | | |
| Genetic reagent (D. melanogaster) | *UAS-myr::GCaMP5a* | This paper | | transgenic flies harboring *UAS-myr::GCaMP5a* |
| Genetic reagent (D. melanogaster) | *UAS-myr::R-GECO1* | This paper | | transgenic flies harboring *UAS-myr::R-GECO1* |
| Genetic reagent (D. melanogaster) | *QUAS-myr::GCaMP5a* | This paper | | transgenic flies harboring *QUAS-myr::GCaMP5a* |
| Genetic reagent (D. melanogaster) | *13XLexAop2-myr ::GCaMP6s* | This paper | | transgenic flies harboring *13XLexAop2-myr::GCaMP6s* |
| Antibody | mouse monoclonal anti-Brp | DSHB | Cat# nc82 | 1:50 |
| Antibody | mouse monoclonal anti-c-Myc | DSHB | Cat# 9E10 | 1:100 |
| Chemical compound, drug | tetrodotoxin | Tocris | Cat# 1078 | 1 µM |
| Chemical compound, drug | lanthanum chloride | Sigma-Aldrich | Cat# 211605 | 0.1 mM, 1 mM |
| Chemical compound, drug | acetylcholine | Sigma-Aldrich | Cat# A6625 | 2.5 mM |
| Chemical compound, drug | γ-aminobutyric acid (GABA) | Sigma-Aldrich | Cat# A2129 | 2.5 mM |
| Chemical compound, drug | glutamate | Sigma-Aldrich | Cat# G1626 | 2.5 mM |
| Chemical compound, drug | tyramine | Sigma-Aldrich | Cat# T90344 | 2.5 mM |
| Chemical compound, drug | octopamine | Sigma-Aldrich | Cat# O0250 | 2.5 mM |
| Chemical compound, drug | N-acetyl cysteine | Sigma-Aldrich | Cat# A7250 | 2.5 mM |
| Chemical compound, drug | hydrogen peroxide | Sigma-Aldrich | Cat# H1009 | 0.1 mM |
| Chemical compound, drug | halocarbon oil 27 | Sigma-Aldrich | Cat# H8773 | |
| Chemical compound, drug | dihydroethidium (DHE) | Sigma-Aldrich | Cat# 309800 | 30 µM |
| Recombinant DNA reagent | pUAST-myr::GCaMP5a | This paper | Fly germline transformation plasmid | GCaMP5a DNA with myristoylation sequence fused at the 5' end |
| Recombinant DNA reagent | pQUAST-myr::GCaMP5a | This paper | Fly germline transformation plasmid | GCaMP5a DNA with myristoylation sequence fused at the 5' end |
| Recombinant DNA reagent | pUAST-myr::R-GECO1 | This paper | Fly germline transformation plasmid | R-GECO1 DNA with myristoylation sequence fused at the 5' end |
| Recombinant DNA reagent | pJFRC19-13XLexAop2-IVS-myr::GCaMP6s | This paper | Fly germline transformation plasmid | GCaMP6s DNA with myristoylation sequence fused at the 5' end |

*Continued on next page*

*Continued*

| Reagent type (species) or resource | Designation | Source or reference | Identifiers | Additional information |
|---|---|---|---|---|
| Software, algorithm | Volocity | PerkinElmer, Inc | http://www.perkinelmer.com/lab-products-and-services/cellular-imaging/performing-advanced-image-data-analysis.html | |
| Software, algorithm | Slidebook | Intelligent Imaging Innovations, Inc | https://www.intelligent-imaging.com/slidebook | |
| Software, algorithm | Fiji | *Schindelin et al., 2012* | https://fiji.sc/ | |
| Software, algorithm | Graphpad Prism 7 | GraphPad software | https://www.graphpad.com/scientific-software/prism/ | |
| Software, algorithm | Igor Pro | WaveMetrics, Inc | https://www.wavemetrics.com/products/igorpro | |
| Software, algorithm | AQuA | *Wang et al., 2019* | https://github.com/yu-lab-vt/AQuA#fiji-plugin; *Wang, 2019* | |

### *Drosophila* stocks and husbandry

All larvae/flies were cultured in cornmeal food at 25 ℃ under 12 hr/12 hr dark/light cycles. Female larvae were used for all experiments unless otherwise stated. The specific developmental stages studied in each experiment are indicated in the following Materials and method details. *Drosophila* strains used include: Bloomington stock center $trpml^1$ (28992), $trpml^2$ (42230), $trpml^{JF01239}$ (31294), $tyrR^{JF01878}$ (25857), $tyrRII^{JF02749}$ (27670), $trpA1^1$ (36342), $trpm^2$ (35527), $nompC^3$ (42258), $trp^1$ (5692), $trpl^{302}$ (31433), $pkd2^1$ (24495), $trp\gamma^{JF01244}$ (31299), $pyx^{JF01242}$ (31297), btl-Gal4 (8807), UAS-Lifeact-GFP (57326), UASp-αTub-GFP (7373), 10XUAS-IVS-myr::tdTomato (32222), UAS-trpml-myc (57372), UAS-GFP-Lamp1 (42714). Oct-TyrR$^{hono}$ (*Nagaya et al., 2002*), $nompC^4$ (*Walker et al., 2000*), pain-less$^{70}$ (*Im et al., 2015*), $nan^{36a}$ (*Kim et al., 2003*), $wtrw^{ex}$ (*Kim et al., 2010*), $Tdc2^{RO54}$ (*Cole et al., 2005*), $T\beta h^{nM18}$ (*Monastirioti et al., 1996*), alrm-Gal4(*Doherty et al., 2009*), alrm >QF > Gal4, repo-FLPase, alrm-LexA::GAD (*Stork et al., 2014*), btl-LexA (*Roy et al., 2014*), 5XUAS-trpml-GCaMP5g (*Wong et al., 2017*). UAS-myr::GCaMP5a, UAS-myr::R-GECO1, QUAS-myr::GCaMP5a, 13XLexAop2-myr::GCaMP6s flies were generated in this study.

### Constructs and transgenic flies

The full-length ORFs of GCaMP5a, R-GECO1, GCaMP6s with an in-frame DNA fragment encoding the myristoylation signal peptide at 5'-end were cloned into vectors pUAST, pQUAST, pJFRC19 (harboring 13XLexAop2-IVS, referring to the plasmid Addgene Cat# 26224) to generate constructs pUAST-myr::GCaMP5a, pQUAST-myr::GCaMP5a, pUAST-myr::R-GECO1, pJFRC19-13XLexAop2-IVS-myr::GCaMP6s for injection. The transgenic flies were injected and recovered by Rainbow Transgenic Flies, Inc (California).

### $Ca^{2+}$ imaging and data analysis

$Ca^{2+}$ imaging in intact larvae: the $1^{st}$ instar larva expressing myr-GCaMP5a in astrocytes (25℃, 24–32 hr after egg laying) was sandwiched in 30 μl halocarbon oil 27 (Cat# H8773, Sigma-Aldrich) between a slide and a 22 × 22 mm coverslip (Cat# 1404–15, Globe Scientific Inc), then a 3 min time-lapse video was taken immediately on a spinning disk confocal microscope equipped with a 40X oil immersion objective.

CNS dissection from early $3^{rd}$ instar larvae expressing myr-GCaMP5a in astrocytes (larval density ~100, 25℃, 76–84 hr after egg laying) was performed in the imaging buffer (pH7.2) containing 110 mM NaCl, 5.4 mM KCl, 0.3 mM $CaCl_2$, 0.8 mM $MgCl_2$, 10 mM D-glucose, 10 mM HEPES, the CNS was immediately transferred to a silicone coated petri dish, immersed in 100 μl imaging buffer (1.2 mM $CaCl_2$), and immobilized gently by sticking the attached nerves onto the silicone surface with forceps. The petri dish then was placed on the stage of a spinning disk confocal microscope equipped with a 40X water dipping objective. The focal plane was fixed around where most

of the dorsal lateral astrocytes start to appear in the field of view. After 4 min acclimation and stabilization, a 6 min movie (excitation channel, 488 nm. exposure time, 300 ms. single focal plane) was taken for analysis. The 488 nm and 516 nm channels were alternated for imaging both the microdomain $Ca^{2+}$ transients in astrocytes and the dynamics of tracheal filopodia. To keep the tracheal filopodia in focus during the course of extension and retraction, images spanning 5 µM in z depth were taken.

The frequency (the number of microdomain $Ca^{2+}$ transients per minute) of microdomain $Ca^{2+}$ transients in each preparation was initially quantified as follows: a 100 µm X 100 µm window was cropped from each movie and resulted in nine smaller, side-to-side 100pixel X 100pixel windows (line drawing over movies) in which the number of microdomain $Ca^{2+}$ transients was counted manually. The total number of microdomain $Ca^{2+}$ transients in each preparation (100 µm X 100 µm) was acquired by adding up all the numbers counted in these 9 100pixel X 100pixel windows. The cutoff for defining a $Ca^{2+}$ transient is 5% change in delta $F/F_0$, which is evaluated by *post hoc* calculations after manually selecting active events. After the automatic $Ca^{2+}$ signal analysis software Astrocyte Quantitative Analysis (AQuA)(*Wang et al., 2019*) became publicly available, we also compared our initial datasets of the dissected CNS from wildtype $3^{rd}$ instar larvae using the above manual method to the ones obtained through AQuA (Manual vs AQuA-Source Data 8), and we didn't find a significant difference (manual: $16.7 \pm 3.69$ versus AQuA: $19.8 \pm 3.02$, p=0.53). The intensity of microdomain $Ca^{2+}$ transients was measured with software Volocity (PerkinElmer, Inc), and the amplitude of microdomain $Ca^{2+}$ transients was defined by $(F_t-F_0)/F_0$ (t = 0,1,2...40, the peak amplitude was designated at t = 20, delta $F = F_t-F_0$) as percentage. The full width at half maximum (FWHM) of astrocyte microdomains was acquired using software Igor Pro (WaveMetrics, Inc).

For bath application of compounds, halfway through the 6 min imaging window (~3 min), 100 µl imaging buffer (1.2 mM $Ca^{2+}$) containing drugs (2X final concentration) was directly applied onto the preparations, then imaging continued for another 3 min. The chemicals used for bath application experiments include: Tocris, tetrodotoxin (1 µM, Cat# 1078). Sigma-Aldrich, lanthanum chloride ($LaCl_3$, Cat# 211605), acetylcholine (2.5 mM, Cat# A6625), γ-aminobutyric acid (GABA, 2.5 mM, Cat# A2129), glutamate (2.5 mM, Cat# G1626), tyramine (2.5 mM, Cat# T90344), octopamine (2.5 mM, Cat# O0250), N-acetyl cysteine (NAC, 2.5 mM, Cat# A7250), hydrogen peroxide ($H_2O_2$, 0.1 mM, Cat# H1009).

## Immunostaining and tracheal branch tracing

The CNS dissected in PBS from $3^{rd}$ instar larvae (larval density ~100, 25℃, 100–108 hr after egg laying) was immediately transferred in 4% formaldehyde for fixation for 20 min (for co-staining with tracheal filopodia, the dissection was performed in the imaging buffer with 0.3 mM $Ca^{2+}$, and the CNS preparations were incubated in the imaging buffer with 1.2 mM $Ca^{2+}$ for 10 min before 4% formaldehyde fixation). Washing in PBS for $3 \times 10$ min. Permeabilization in PBS + 0.3% Triton X-100 for 2 hr. Primary antibody (1:50 anti-Brp, DSHB, Cat# nc82 in PBS + 0.1% Triton X-100) incubation at 4 ℃ for ~72 hr. Secondary antibody incubation at room temperature for ~2 hr. Tracheal branches in ventral nerve cord were illuminated with 408 nm laser light and emitting autofluorescence was imaged. Each individual branch was then traced manually with Simple Neurite Tracer (Fiji).

## Reactive oxygen species detection by DHE (dihydroethidium) staining

The CNS preparations from $3^{rd}$ instar larvae (larval density ~100, 25℃, 100–108 hr after egg laying) were made exactly in the same way for $Ca^{2+}$ imaging. Incubation in 100 µl imaging buffer (1.2 mM $Ca^{2+}$) containing 30 µM DHE for 8 min before imaging (30 µm in z depth, starting from the very dorsal side, was taken). The puncta were automatically counted by the segmentation tool in Slidebook.

### Statistics

All statistics were performed in Graphpad. No data were excluded for analyses. 2–3 replications were successfully performed for each experiment. Comparison between groups was tested by one-way ANOVA with Tukey's *post hoc* tests, or unpaired t-test. Comparison within groups was tested by paired t-test. p<0.05 was considered statistically significant. *p<0.05, **p<0.01.

## Data availability

Source data for *Figure 1C–E*; *Figure 2A–H*; *Figure 3E and F*; *Figure 4A–D*; *Figure 2—figure supplement 1 – Figure 2A–C and E*; *Figure 3—figure supplement 1 – Figure 3D*; *Figure 4—figure supplement 1 - Figure 4A–B and D* are included. Materials generated for this study will be freely available on request.

## Acknowledgements

We thank Bloomington Stock Center, Kyoto Stock Center, Drs. CS Zuker, TB Kornberg, C Montell, J Hirsh and M Monastirioti for providing flies. We thank A Sheehan for making constructs. We thank Freeman lab members for feedback on the manuscript. This work was supported by NIH RO1 NS053538 (to MRF) and OHSU.

## Additional information

### Funding

| Funder | Grant reference number | Author |
|---|---|---|
| National Institutes of Health | RO1 NS03538 | Marc Freeman |
| Oregon Health and Science University | NIH RO1 NS053538 | Marc Freeman |

The funders had no role in study design, data collection and interpretation, or the decision to submit the work for publication.

### Author contributions

Zhiguo Ma, Conceptualization, Data curation, Formal analysis, Validation, Visualization, Methodology, Writing - original draft, Writing - review and editing; Marc R Freeman, Conceptualization, Resources, Supervision, Funding acquisition, Writing - original draft, Project administration, Writing - review and editing

### Author ORCIDs

Zhiguo Ma https://orcid.org/0000-0003-2250-1861
Marc R Freeman https://orcid.org/0000-0003-3481-3715

### Decision letter and Author response

Decision letter https://doi.org/10.7554/eLife.58952.sa1
Author response https://doi.org/10.7554/eLife.58952.sa2

## Additional files

### Supplementary files

• Source data 1. Manual VS AQuA. Manual method and AQuA identify comparable amount of microdomain $Ca^{2+}$ transients.

• Transparent reporting form

### Data availability

All data generated or analyzed during this study are included in the manuscript and supporting files.

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
