## [Decision Letter]

**Acceptance summary:**

The manuscript by Ma and Freeman describes a role for the TRP channel *trpml* in regulating astrocyte microdomain calcium oscillations in *Drosophila*. They link this biology to regulation of local tracheal filopodial dynamics and ROS activity, suggesting one role for astrocyte calcium dynamics is to control local gas exchange in *Drosophila* larvae. This study represents a significant advance in understanding how microdomain calcium transients are generated and regulate CNS physiology and function.

**Decision letter after peer review:**

Thank you for submitting your article "TrpML-mediated astrocyte microdomain Ca 2+ transients regulate astrocyte-tracheal interactions" for consideration by *eLife*. Your article has been reviewed by three peer reviewers, and the evaluation has been overseen by a Reviewing Editor and Ronald Calabrese as the Senior Editor. The following individual involved in review of your submission has agreed to reveal their identity: Brian A MacVicar (Reviewer #2).

The reviewers have discussed the reviews with one another and the Reviewing Editor has drafted this decision to help you prepare a revised submission.

Summary:

Ma and Freeman present results from studies of astrocyte Ca^2+^ transients in the *Drosophila* nervous system and their relationship to the physiology and behavior of CNS trachea. Using genetic Ca^2+^ reporters, they first show that astrocyte processes exhibit transient changes in Ca^2+^ concentration within microdomains. They present results suggesting that extracellular Ca^2+^ is required for Ca^2+^ changes within microdomains, and they identify the TrpML ion channel, in a targeted genetic screen, as a critical component of the mechanism regulating Ca^2+^ within microdomains. Importantly, Ca^2+^ transients are almost eliminated with astrocyte-specific knockdown of *trpml^1^*. The authors further show that tyramine, acting through a fly tyramine receptor (TyrRII), can enhance Ca^2+^ transients but that it is not required for such transients. They demonstrate that the Ca^2+^ microdomains are in close proximity to filopodia of tracheal tubules, and go on to show that transient changes in Ca^2+^ within such microdomains can influence the behavior of tracheal filopodia.

Their results suggest that increased Ca^2+^ within such microdomains is strongly correlated with filapodial retraction events. Additional findings show that extension of tracheal filopodia is enhanced in the *trpml^1^* mutant, whereas retraction events are normal, suggesting that the TrpML channel is important for inhibition of extension events. Finally, the authors show that there is overgrowth of trachea and higher levels of reactive oxygen species in the *trpml^1^* mutant, suggesting a role of the TrpML channel and Ca^2+^ in tracheal growth and gas exchange.

The manuscript is clearly written and represents a significant advance in understanding Ca^2+^ regulation in astrocytes. The results are important as similar astrocyte Ca^2+^ microdomains regulate CNS physiology in mammals, and *Drosophila* provides an important genetic model for the analysis of astrocyte Ca^2+^ signaling. However, there are a few issues and suggestions for improvement, as well as points that need clarification prior to publication.

Essential revisions:

1) Based on the data presented, it is not clear whether the TRP channel is the primary source of calcium influx for microdomain oscillations, or secondarily coupled to the process to control the frequency of the events only. Please clarify this or provide more convincing data that *trpml* directly mediates calcium microdomain influx or tone down conclusion and add more to the Discussion that discusses alternative models.

2) Where is *trpml* localized in *Drosophila* astrocytes? All reviewers agree that it is important to know whether TrpML is acting in astrocyte lysosomes or at the cell membrane to regulate CNS and tracheal functions. Localizing it would help, as would measuring the amount of calcium influx signal in the absence of the channel --- the frequency goes down, but they still appear to be present. See reviewer comments for specific comments and suggestions.

3) Please address whether astrocytes are regulating trachea in the mature larval CNS, as detailed in comments by reviewer 3. Alternatively, if the alteration of tracheal function is a consequence of abnormal astrocyte development.

4) The comparisons made between the results in *Drosophila* and mouse astrocytes need to be clarified in some cases, and the authors should be more careful in discussing the different types of mouse and rat astrocyte calcium signals (see specific comments by reviewer 2)

Reviewer #1:

The manuscript of Ma and Freeman is clearly written and represents a significant advance in understanding Ca^2+^ regulation in astrocytes. The results are important as similar astrocyte Ca^2+^ microdomains regulate CNS physiology in mammals, and *Drosophila* provides an important genetic model for the analysis of astrocyte Ca^2+^ signaling.

The major issues detailed below should be addressed prior to publication:

1) The authors show that elimination of extracellular and the TrpML channel inhibit Ca^2+^ transients within astrocyte microdomains. The presentation of these results leads the reader to think that TrpML acts at the astrocyte cellular membrane to regulate Ca^2+^ entry. While this might be true, The TrpML channel has been described as a lysosomal membrane rather than a cellular membrane channel, and it is clearly required for cell migration and phagocytosis via regulation of actomyosin contractility (Edwards-Jorquera et al., JCB, 2020; https://doi.org/10.1083/jcb.201905228). Is there any evidence that TrpML is even present at the cellular membrane? Have the authors determined localization of the channel in astrocytes? If not, the presentation will lead most readers to think its site of action is the cellular membrane when this has not been demonstrated and is likely not to be the case. A more detailed model for TrpML, taking its localization into account, is needed.

2) The authors propose "that one important physiological role for astrocyte microdomain Ca^2+^ transients is to modulate CNS gas exchange through TrpML and ROS signaling." This suggests that TrpML might have a modulatory role in the mature nervous system. However, it seems that all of the authors' results about TrpML could be interpreted as a requirement for the channel in astrocytes during development with a consequent effect on physiology of the tracheal system. Indeed, the authors show that tracheal overgrowth occurs in the channel mutant. I wonder if there is any evidence for a role of TrpML in the mature nervous system. Did the authors attempt a conditional knockdown of TrpML in the mature larval CNS? What about inhibition of TrpML in mature CNS astrocytes using an antagonist that blocks Ca^2+^ release from lysosomes? Or, activation of TrpML with an agonist in the mature CNS? Or, transient blockade of Ca^2+^ entry in the mature larval CNS? Is it known whether such manipulations alter tracheal function? It would seem important to determine whether astrocytes actually modulate trachea and/or gas exchange in the mature CNS since this is the implication of the authors' model.

3) Although TrpML is known to be Ca^2+^ permeable, it is also permeable to other cations. It is described in FlyBase as a cation channel involved in regulation of endosomal trafficking, autophagy, lateral inhibition and TOR signaling in hemocytes and perhaps other cell types. Is regulation of astrocytes and tracheal function as simple as modulation of Ca^2+^ entry or is it more complicated than that? I wonder if Ca^2+^ microdomains are affected indirectly because of an effect on endosomal trafficking. Some discussion of this is needed.

4) There is no mention of CNS abnormalities in the *trpml^1^* mutant or *trpml^1^* knockdown flies. Does the CNS appear to be completely normal with absence of TrpML?

Reviewer #2:

This is very interesting paper that provides convincing data that calcium microdomains in *Drosophila* astrocytes occur, are triggered by reactive oxygen species (ROS), are enhanced in frequency by the neurotransmitter tyramine via the TyrRII receptor and that they decrease in frequency by ~70% to 80% in *trpml* loss of function mutants. The authors also show that tracheal growth is modified by these calcium signals in astrocytes. These data provide intriguing counterparts to observations in mammalian astrocytes with respect to calcium signaling. However, there are comparisons that the authors make between the results in *Drosophila* and mouse astrocytes that need to be clarified in some cases and the authors should be more careful in discussing the different types of mouse and rat astrocyte calcium signals. Finally, the TRPML1 channel is expressed on lysosomes according to the report they cite from Zhang et al., 2016. The results presented here indicate expression on the plasma membrane based on the block by lanthanum and effects of removing external calcium. This puzzle and apparent discrepancy should be addressed by the authors.

1) "While knockout of 11 of these TRP channels had no effect, we found that microdomain Ca^2+^ events decrease by ~70% to 80% in *trpml* loss of function mutants, in both intact 1st instar larvae (Figure 1—figure supplement 1E) and acute CNS preparations from 3rd instar larvae (Figure 2D; –Figure 2—figure supplement 2—video 1".

This is a key finding of this study and this careful analysis shows the strength and power of working with *Drosophila* as a model system.

2) "Astrocyte microdomain Ca^2+^ transients in *Drosophila* share many features with those observed in mammals(Agarwal et al., 2017; Nett et al., 2002; Srinivasan et al., 2015)."

The fly astrocyte microdomains have overlap with the observations more recently in mammalian astrocytes but the authors should be careful to differentiate between the older observations of IP3-mediated calcium release from ER (e.g. Nett et al., 2002) and the observations of calcium entry dependent microdomains. I don't think the microdomains were observed using the older dye loading techniques. For example, the Nett et al. paper showed almost complete block of the astrocyte calcium signals by the IP3-R antagonist heparin. In my opinion these fly microdomains that are described here are most similar to the microdomains first reported by Srinivasan et al., 2015. Rungta et al., 2016, supports Srinivasan et al. and their data also point to a calcium entry pathway independent of IP3R mediated release and not altered in TRPA1-/- mice. Of course, there appear to be multiple sources of calcium that generate microdomains and Agarwal et al., 2017, point to a mechanism of calcium release from mitochondria. The differences between these various mechanisms generating calcium transients should be clearly stated because different types of calcium signals generated in different regions of astrocytes may potentially have different functions (as would be expected in neurons).

3) This observation of ROS sensitivity in this study is consistent with the report by Zhang et al., 2016, describing the ROS sensitivity of TRPMl1 channels. However, there is a significant difference in that Zhang et al. describe the calcium signals as arising from the ROS triggered release of calcium from lysosomes into the cytoplasm not as a plasma membrane channel leading to entry from the extracellular space. The authors should reconcile this important difference and describe which signals are lysosomal release dependent.

4) "We note that those that exhibited only retraction did so very early in the imaging window, which could indicate that we began our imaging after extension had been initiated."

This paragraph is not critical but is confusing. It's clear that 6 min is sufficient to observe a cycle of extension -retraction but what is surprising is that retractions alone occur at a higher rate than extensions alone. Wouldn't these be the same if the probability was similar? Alternatively could imaging the tissue may make retractions much more likely perhaps by generating ROS itself from the tissue illumination in the spinning disk confocal?

5) "Together, these observations indicate that a large fraction of astrocyte

microdomain Ca^2+^ transients are spatiotemporally correlated with the retraction of

adjacent tracheal filopodia."

This conclusion seems reasonable based on the imaging data.

Reviewer #3:

The manuscript by Ma and Freeman describes a role for the TRP channel *trpml* in regulating astrocyte microdomain calcium oscillations in *Drosophila*. They link this biology to regulation of local tracheal filopodial dynamics and ROS activity, suggesting one role for astrocyte calcium dynamics is to control local gas exchange in *Drosophila* larvae. The authors' data set is quite interesting and relevant to the field. I have a few suggestions for improvement and some general clarification points as well.

1) The authors imply that the *trpml* channel directly mediates microdomain calcium influx in astrocytes, but I don't think their data separates a role for the channel being the source of calcium influx or secondarily coupled to the process to control the frequency of the events only. As such, the authors need to state this or provide more convincing data that *trpml* directly mediates calcium microdomain influx.

2) Related to the point above, the authors should document and show the amplitude of calcium transients in *trpml* knockdowns and mutants as they do for wildtype in Figure 1B. Is *trpml* required only for the frequency of these events, or also the amplitude. The authors must have this data already in hand. If the amplitude of the residual events in the same, it seems more likely that *trpml* is controlling when these events occur, rather than being the direct channel for gating the calcium events themselves.

3) Similarly, where is *trpml* localized in *Drosophila* astrocytes – is it on the plasma membrane at sites of calcium influx? If it is located on lysosomes or internal compartments, I have a hard time understanding how it could be directly mediating calcium influx from the extracellular space.

4) It would be helpful to use the *trpml* RNAi line from Figure 2D and Figure 4B for the retraction/extension assays in Figure 4A. In 4A, only the *trpml* mutant is used to show defects in tracheal filopodia. If this biology requires *trpml* in astrocytes, knocking it down only in these cells should trigger this phenotype. Alternatively, this biology could be coming from an independent role of *trpml* in tracheal cells. This experiment is important, especially given how small some of the reported changes appear to be. Since the authors used *trpml* RNAi knockdown in astrocytes for all there other manipulations, it raises a question as to why not here in one of the most important experiments.

5) The most difficult part of the paper for me is Figure 3D. I have a hard time finding compelling images here to support the model. From the bottom astrocyte GCaMP traces (red) and tracheal filopodial length (red) traces, only pair 2 looks compelling – red goes up, green goes down. The rest are bouncing around all over the place and it is hard to see a compelling correlation. Would it be possible to add a supplementary figure showing more of just these traces for all (or a larger representative sample) of the data included in Figure 3E. That figure shows a perfect correlation, but the raw data provided do not look convincing. Seeing more of those raw data would be very helpful for convincing the readers of this correlation.

---

## [Author Response]

Essential revisions:1) Based on the data presented, it is not clear whether the TRP channel is the primary source of calcium influx for microdomain oscillations, or secondarily coupled to the process to control the frequency of the events only. Please clarify this or provide more convincing data that trpml directly mediates calcium microdomain influx or tone down conclusion and add more to the Discussion that discusses alternative models.

We agree that TrpML may be the channel, or coupled to the channel that drive the majority of calcium influx. Given that we cannot test this directly, we have toned down these conclusions and added a discussion of these alternative models to the Discussion.

2) Where is trpml localized in Drosophila astrocytes? All reviewers agree that it is important to know whether TrpML is acting in astrocyte lysosomes or at the cell membrane to regulate CNS and tracheal functions. Localizing it would help, as would measuring the amount of calcium influx signal in the absence of the channel --- the frequency goes down, but they still appear to be present. See reviewer comments for specific comments and suggestions.

This is an interesting question that we have addressed by assaying localization of TrpML in astrocytes. In new data (Figure 2—figure supplement 1F) we show that TrpML-MYC is localized (1) near the membrane of astrocyte cell bodies, and appears to be sub-cortical based on co-staining with plasma membrane-tethered GFP; and (2) in a punctate pattern throughout astrocytes, which largely overlaps with the endo-lysosomal marker GFP-Lamp1.

3) Please address whether astrocytes are regulating trachea in the mature larval CNS, as detailed in comments by reviewer 3. Alternatively, if the alteration of tracheal function is a consequence of abnormal astrocyte development.

Astrocyte development based on the analysis of a number of markers and cellular morphology appears normal in TrpML mutants (e.g. Figure 2—figure supplement 1D). We have no evidence that loss of TrpML (in mutants or RNAi lines) results in developmental defects in astrocytes. Astrocyte modulation of trachea occurs at all larval stages tested, including the 3^rd^ larval instar. For instance, tracheal morphologies were quantified in Figure 4—figure supplement 1B in 3^rd^ instar larvae.

4) The comparisons made between the results in Drosophila and mouse astrocytes need to be clarified in some cases, and the authors should be more careful in discussing the different types of mouse and rat astrocyte calcium signals (see specific comments by reviewer 2)

We have clarified this point throughout the manuscript according to reviewer #2’s request.

Reviewer #1:The manuscript of Ma and Freeman is clearly written and represents a significant advance in understanding Ca^2+^ regulation in astrocytes. The results are important as similar astrocyte Ca^2+^ microdomains regulate CNS physiology in mammals, and Drosophila provides an important genetic model for the analysis of astrocyte Ca^2+^ signaling.The major issues detailed below should be addressed prior to publication:1) The authors show that elimination of extracellular and the TrpML channel inhibit Ca^2+^ transients within astrocyte microdomains. The presentation of these results leads the reader to think that TrpML acts at the astrocyte cellular membrane to regulate Ca^2+^ entry. While this might be true, The TrpML channel has been described as a lysosomal membrane rather than a cellular membrane channel, and it is clearly required for cell migration and phagocytosis via regulation of actomyosin contractility (Edwards-Jorquera et al., JCB, 2020; https://doi.org/10.1083/jcb.201905228). Is there any evidence that TrpML is even present at the cellular membrane? Have the authors determined localization of the channel in astrocytes? If not, the presentation will lead most readers to think its site of action is the cellular membrane when this has not been demonstrated and is likely not to be the case. A more detailed model for TrpML, taking its localization into account, is needed.

In the revised manuscript we have added data to address this directly by assaying localization of TrpML in astrocytes. In new data (Figure 2—figure supplement 1F) we show that TrpML-MYC is localized (1) near the membrane of astrocyte cell bodies, and appears to be sub-cortical based on co-staining with membrane-tethered GFP; and (2) in a punctate pattern throughout astrocytes, which largely overlaps with the endo-lysosomal marker GFP-Lamp1. As such, both possibilities exist—that TrpML is functioning near/at the membrane or at endo-lysosomes. We have included this possibility in our model for proposed TrpML function.

2) The authors propose "that one important physiological role for astrocyte microdomain Ca^2+^ transients is to modulate CNS gas exchange through TrpML and ROS signaling." This suggests that TrpML might have a modulatory role in the mature nervous system. However, it seems that all of the authors' results about TrpML could be interpreted as a requirement for the channel in astrocytes during development with a consequent effect on physiology of the tracheal system. Indeed, the authors show that tracheal overgrowth occurs in the channel mutant. I wonder if there is any evidence for a role of TrpML in the mature nervous system. Did the authors attempt a conditional knockdown of TrpML in the mature larval CNS? What about inhibition of TrpML in mature CNS astrocytes using an antagonist that blocks Ca^2+^ release from lysosomes? Or, activation of TrpML with an agonist in the mature CNS? Or, transient blockade of Ca^2+^ entry in the mature larval CNS? Is it known whether such manipulations alter tracheal function? It would seem important to determine whether astrocytes actually modulate trachea and/or gas exchange in the mature CNS since this is the implication of the authors' model.

Our assumption is that by modulating tracheal growth, astrocytes can thereby regulate gas exchange by increasing or decreasing tracheal filopodial coverage of the CNS, since trachea is the primary route of gas exchange in the brain. Our argument is based on the increased ROS in *trpml* mutants or when *trpml* is depleted from astrocytes (Figure 4D). Our best evidence for active regulation of tracheal elements by astrocytes in the mature (3^rd^ instar) larval nervous system is that bath application of tyramine, which activates microdomain transients, drives filopodial retraction in live preparations (Figure 4B). We included our new data in the revised manuscript that showed *trpml* knockdown specifically at the 3^rd^ instar larval stage did result in faster tracheal filopodial growth. Interestingly, such knockdown in trachea also make trachea filopodia grow more rapidly. This suggests that *trpml* function both in astrocytes and trachea controlling tracheal filopodial dynamics through increasing growth rate. Finally, All of the approaches suggested above to manipulate calcium would likely work in live preparations (e.g. Figure 1D, LaCl_3_ and 0mM extracellular calcium) for blockade of different aspects of calcium signaling in astrocytes, but it would not spare trachea from influence, and more importantly not be technically feasible for us to do this chronically, allow animals to grow, and then observe how this alters tracheal morphology. Nevertheless, in response to this concern we will temper our conclusions regarding active regulation of gas exchange and leave open alternative possibilities for the phenotypes we observe.

3) Although TrpML is known to be Ca^2+^ permeable, it is also permeable to other cations. It is described in FlyBase as a cation channel involved in regulation of endosomal trafficking, autophagy, lateral inhibition and TOR signaling in hemocytes and perhaps other cell types. Is regulation of astrocytes and tracheal function as simple as modulation of Ca^2+^ entry or is it more complicated than that? I wonder if Ca^2+^ microdomains are affected indirectly because of an effect on endosomal trafficking. Some discussion of this is needed.

The reviewer is correct, TrpML may exert its effects through other ions, or by moving multiple ions. It may also be more complicated than the model we propose, and we have discussed these points in the revised manuscript. We do not think there is a general disruption of calcium signaling based on the following evidence: (1) in new data we show that the amplitude of calcium transients in astrocytes in not altered in *trpml* mutants, it is the frequency that changes (Figure 2—figure supplement 1B); (2) tyramine stimulates normal somatic calcium signal in *trpml* mutant astrocytes. (Figure 2—figure supplement 1B); (3) astrocyte development even in *trpml* mutants appears normal, and one might expect developmental defects if endosomal trafficking was significantly perturbed.

4) There is no mention of CNS abnormalities in the trpml^1^ mutant or trpml^1^ knockdown flies. Does the CNS appear to be completely normal with absence of TrpML?

Overall the CNS appears grossly normal based on size and the morphology of glia, and major tracheal branches (minor overgrowth phenotypes). However, we note that some phenotypes in *trpml* do mimic mucolipidosis type IV, in that we can visualize autofluorescent material in the CNS. We also note (also below) that the lethality of *trpml* mutants is a result of defective astrocyte function—we are able to rescue ~100% of *trpml* mutants to adulthood by astrocyte-specific expression of TrpML (Figure 2—figure supplement 1E).

Reviewer #2:This is very interesting paper that provides convincing data that calcium microdomains in Drosophila astrocytes occur, are triggered by reactive oxygen species (ROS), are enhanced in frequency by the neurotransmitter tyramine via the TyrRII receptor and that they decrease in frequency by ~70% to 80% in trpml loss of function mutants. The authors also show that tracheal growth is modified by these calcium signals in astrocytes. These data provide intriguing counterparts to observations in mammalian astrocytes with respect to calcium signaling. However, there are comparisons that the authors make between the results in Drosophila and mouse astrocytes that need to be clarified in some cases and the authors should be more careful in discussing the different types of mouse and rat astrocyte calcium signals.

Thank you for making this point. We have tried to clarify this comparison in the revised manuscript to be accurate and fair. We welcome any additional feedback from the reviewer on clarification if appropriate..

Finally, the TRPML1 channel is expressed on lysosomes according to the report they cite from Zhang et al., 2016. The results presented here indicate expression on the plasma membrane based on the block by lanthanum and effects of removing external calcium. This puzzle and apparent discrepancy should be addressed by the authors.

In the revised manuscript we have added data to address this directly by assaying localization of TrpML in astrocytes. In new data (Figure 2—figure supplement 1F) we show that TrpML-MYC is localized (1) near the membrane of astrocyte cell bodies, and appears to be sub-cortical based on co-staining with membrane-tethered GFP; and (2) in a punctate pattern throughout astrocytes, which largely overlaps with the endo-lysosomal marker GFP-Lamp1. As such, both possibilities exist—that TrpML is functioning near/at the membrane or at endo-lysosomes. We have included this possibility in our model for proposed TrpML function.

1) "While knockout of 11 of these TRP channels had no effect, we found that microdomain Ca^2+^ events decrease by ~70% to 80% in trpml loss of function mutants, in both intact 1st instar larvae (Figure 1—figure supplement 1E) and acute CNS preparations from 3rd instar larvae (Figure 2D; Figure 2—figure supplement 2—video 1".This is a key finding of this study and this careful analysis shows the strength and power of working with Drosophila as a model system.

We thank the reviewer for the enthusiasm and support for the model.

2) "Astrocyte microdomain Ca^2+^ transients in Drosophila share many features with those observed in mammals(Agarwal et al., 2017; Nett et al., 2002; Srinivasan et al.,2015)."The fly astrocyte microdomains have overlap with the observations more recently in mammalian astrocytes but the authors should be careful to differentiate between the older observations of IP3-mediated calcium release from ER (e.g. Nett et al., 2002) and the observations of calcium entry dependent microdomains. I don't think the microdomains were observed using the older dye loading techniques. For example, the Nett et al. paper showed almost complete block of the astrocyte calcium signals by the IP3-R antagonist heparin. In my opinion these fly microdomains that are described here are most similar to the microdomains first reported by Srinivasan et al., 2015. Rungta et al., 2016, supports Srinivasan et al. and their data also point to a calcium entry pathway independent of IP3R mediated release and not altered in TRPA1-/- mice. Of course, there appear to be multiple sources of calcium that generate microdomains and Agarwal et al., 2017, point to a mechanism of calcium release from mitochondria. The differences between these various mechanisms generating calcium transients should be clearly stated because different types of calcium signals generated in different regions of astrocytes may potentially have different functions (as would be expected in neurons).

This is an excellent point raised by the reviewer. In the revised manuscript we have more carefully discussed the diversity of types of calcium signals, and focused primarily on comparison to Agarwal (Agarwal et al., 2017).

3) This observation of ROS sensitivity in this study is consistent with the report by Zhang et al., 2016, describing the ROS sensitivity of TRPMl1 channels. However, there is a significant difference in that Zhang et al. describe the calcium signals as arising from the ROS triggered release of calcium from lysosomes into the cytoplasm not as a plasma membrane channel leading to entry from the extracellular space. The authors should reconcile this important difference and describe which signals are lysosomal release dependent.

Based on the comments of several reviewers we have now included localization studies for TrpML (see above). Based on its localization we believe it could be working by lysosomal release, or by release from the sub-cortical region of the plasma membrane. At the moment we cannot resolve which compartment is the crucial one for astrocyte signaling, so we have left open both possibilities in our model.

For the purposes of informing reviewers, we have attempted to address this experimentally. We did this by examining Ca^2+^ transients using *trpml-GCaMP5G* (which should reflects membrane and lysosomal Ca^2+^ signaling via TrpML) and myr-R-GECO1 (which should be limited to the plasma membrane) (Author response image 1). In many cases observe an overlapping signal between GCaMP and R-GECO-1, implying that at least some TrpML signaling in astrocytes occurs at the plasma membrane.

**Author response image 1. sa2fig1:** Live imaging of TrpML-GCaMP5G and myr-R-GECO1 in astrocytes in 3^rd^ instar larval brain. Note the overlap of signals (dotted circle) arguing for some level of TrpML signaling at the membrane.

4) "We note that those that exhibited only retraction did so very early in the imaging window, which could indicate that we began our imaging after extension had been initiated."This paragraph is not critical but is confusing. It's clear that 6 min is sufficient to observe a cycle of extension -retraction but what is surprising is that retractions alone occur at a higher rate than extensions alone. Wouldn't these be the same if the probability was similar? Alternatively could imaging the tissue may make retractions much more likely perhaps by generating ROS itself from the tissue illumination in the spinning disk confocal?

The reviewer is correct in their assumption. We did indeed observe a statistically equal number of extension and retraction events (Figure 4B) throughout our 6-min imaging window. We apologize for our unclear description and have tried to clarify this. The point we were trying to make is that there are very few filopodia that only retract, and usually they appear at the very beginning of imaging.

5) "Together, these observations indicate that a large fraction of astrocytemicrodomain Ca^2+^ transients are spatiotemporally correlated with the retraction ofadjacent tracheal filopodia."This conclusion seems reasonable based on the imaging data.Reviewer #3:The manuscript by Ma and Freeman describes a role for the TRP channel trpml in regulating astrocyte microdomain calcium oscillations in Drosophila. They link this biology to regulation of local tracheal filopodial dynamics and ROS activity, suggesting one role for astrocyte calcium dynamics is to control local gas exchange in Drosophila larvae. The authors' data set is quite interesting and relevant to the field. I have a few suggestions for improvement and some general clarification points as well.1) The authors imply that the trpml channel directly mediates microdomain calcium influx in astrocytes, but I don't think their data separates a role for the channel being the source of calcium influx or secondarily coupled to the process to control the frequency of the events only. As such, the authors need to state this or provide more convincing data that trpml directly mediates calcium microdomain influx.

We agree with this point, which was also raised by another reviewer, and have clarified this in our revised manuscript.

2) Related to the point above, the authors should document and show the amplitude of calcium transients in trpml knockdowns and mutants as they do for wildtype in Figure 1B. Is trpml required only for the frequency of these events, or also the amplitude. The authors must have this data already in hand. If the amplitude of the residual events in the same, it seems more likely that trpml is controlling when these events occur, rather than being the direct channel for gating the calcium events themselves.

We thank the reviewer for pointing this out. In the revised manuscript we show that the amplitude of calcium transients in control and *trpml* mutants are not significantly different (Figure 2—figure supplement 1B).

3) Similarly, where is trpml localized in Drosophila astrocytes – is it on the plasma membrane at sites of calcium influx? If it is located on lysosomes or internal compartments, I have a hard time understanding how it could be directly mediating calcium influx from the extracellular space.

In the revised manuscript we have added data to address this directly by assaying localization of TrpML in astrocytes. In new data (Figure 2—figure supplement 1F) we show that TrpML-MYC is localized (1) near the membrane of astrocyte cell bodies, and appears to be sub-cortical based on co-staining with membrane-tethered GFP; and (2) in a punctate pattern throughout astrocytes, which largely overlaps with the endo-lysosomal marker GFP-Lamp1. As such, both possibilities exist—that TrpML is functioning near/at the membrane or at endo-lysosomes. We have included this possibility in our model for proposed TrpML function.

4) It would be helpful to use the trpml RNAi line from Figure 2D and Figure 4B for the retraction/extension assays in Figure 4A. In 4A, only the trpml mutant is used to show defects in tracheal filopodia. If this biology requires trpml in astrocytes, knocking it down only in these cells should trigger this phenotype. Alternatively, this biology could be coming from an independent role of trpml in tracheal cells. This experiment is important, especially given how small some of the reported changes appear to be. Since the authors used trpml RNAi knockdown in astrocytes for all there other manipulations, it raises a question as to why not here in one of the most important experiments.

This is an excellent point. We have done these experiments and the data are now included in Figure 4—figure supplement 1A. We found that astrocyte knockdown increased extension rate similar to the *trpml* mutants. These data indicate that astrocyte TrpML can indeed regulate tracheal growth. It seems that tracheal TrpML can also influence growth to some extent. The full effect in the *trpml* mutant is therefore likely due to additivity between astrocytic and tracheal TrpML.

5) The most difficult part of the paper for me is Figure 3D. I have a hard time finding compelling images here to support the model. From the bottom astrocyte GCaMP traces (red) and tracheal filopodial length (red) traces, only pair 2 looks compelling – red goes up, green goes down. The rest are bouncing around all over the place and it is hard to see a compelling correlation. Would it be possible to add a supplementary figure showing more of just these traces for all (or a larger representative sample) of the data included in Figure 3E. That figure shows a perfect correlation, but the raw data provided do not look convincing. Seeing more of those raw data would be very helpful for convincing the readers of this correlation.

We have included an additional set of traces and video frames.